# Evidence for a constant occipital spotlight of attention using MVPA on EEG data

**María Melcón**[1,2]*, **Sander van Bree**[2,3,4], **Yolanda Sánchez-Carro**[5],
**Laura Barreiro-Fernández**[1], **Luca D. Kolibius**[2,3,6], **Elisabet Alzueta**[1,7], **Maria Wimber**[2,3],
**Almudena Capilla**[1], **Simon Hanslmayr**[2,3]

1 Department of Biological and Health Psychology, Universidad Autónoma de Madrid, Madrid, Spain,
2 Centre for Cognitive Neuroimaging, School of Psychology and Neuroscience, University of Glasgow,
Glasgow, United Kingdom, 3 Centre for Human Brain Health, University of Birmingham, Birmingham,
United Kingdom, 4 Department of Medicine, Justus Liebig University, Giessen, Germany, 5 Faculty of
Psychology, European University of the Canary Islands, Santa Cruz de Tenerife, Spain, 6 Department
of Biomedical Engineering, Columbia University, New York, New York, United States of America,
7 Biosciences Division, Center for Health Sciences, SRI International, Menlo Park, California, United
States of America.

* maria.melcon.martin@gmail.com

org/10.1371/journal.pone.0320233

de Paris: Inria Centre de Recherche de Paris,
FRANCE

**Peer Review History:** PLOS recognizes the
benefits of transparency in the peer review
process; therefore, we enable the publication
of all of the content of peer review and
author responses alongside final, published
articles. The editorial history of this article is
available here: https://doi.org/10.1371/journal.
pone.0320233

## Abstract

While traditional behavioural and electroencephalographic studies claim that visuospatial
attention stays fixed at one location at a time, recent research has rather shown that atten-
tion rhythmically fluctuates between locations at different rates. However, little is known
about the temporal dynamics of this fluctuation and whether it changes over time. We
addressed this question by investigating how the neural pattern of visuospatial attention
behaves over space and time by employing classification and conventional analysis of
occipito-parietal EEG activity. Furthermore, we simulated data with the attentional elec-
trophysiological correlates to control for the ground truth that would give rise to certain
classification patterns. We analysed two visuospatial cueing tasks, with a peripheral and a
central cue to control for sensory-driven processes, where attention was covertly oriented
to the left or right hemifield. First, to decode the spatial locus of attention from neural
activity, we trained and tested a classifier on every timepoint from the attentional cue to
the stimulus onset. This resulted in one temporal generalization matrix per participant,
which was time-frequency decomposed to identify the sampling rhythm. Independently,
we calculated a lateralization index based on ERPs and alpha-band power and correlated
these indices with classifier performance. Eventually, we simulated two dataset, with ERPs
and alpha-band attentional modulations, and employed the same decoding approach.
Our results show that attention settled on the cued hemifield in a late time window, but an
early and rhythmic sampling of both hemifields exclusively after the peripheral cue. Only
the ERP lateralization index correlated with classifier performance in the periperhal cue
dataset, suggesting that the early rhythmic state did not reflect attentional sampling but
instead was driven by the cue location, idea also supported by our simulations. Together,
our results characterise the occipital attentional sampling as a constant process slightly
delayed after the cue.

**Data availability statement:** The data and scripts to reproduce the analysis and figures in this paper are available on the Open Science Framework platform (https://osf.io/8e9c7/).

**Funding:** This work was supported by FEDER/Ministerio de Ciencia, Innovación y Universidades – Agencia Estatal de Investigación, Spain (grant PGC2018-100682-B-I00) and the Research Grant FPI-UAM 2017 - Formación de Personal Investigador de la Universidad Autónoma de Madrid (UAM, Spain). None of the funders play any role in the study.

**Competing interests:** The authors have declared that no competing interests exist.

## 1. Introduction

Attention operates as a spotlight, focusing on a single region of space at a time [1–3]. Traditionally, its temporal evolution has been studied through event-related potentials (ERPs) and time-frequency analysis (TF), revealing two strong attentional correlates: the P1/N1 components and alpha-band power [4–9]. Specifically, attentional deployment is associated with an increase in P1/N1 amplitudes and a decrease in alpha power over contralateral occipital regions. Within this framework, attention has been described as a constant process [10]. In other words, when a stimulus appears, the attentional spotlight (AS) is thought to be deployed to its spatial location, remaining there until it is captured again or voluntarily displaced. However, this idea has been challenged by subsequent findings employing paradigms where subjects monitored multiple locations which demonstrated that attention is not static but instead rhythmically fluctuates between locations [11–13, but see 14 for critical considerations], which has been linked to brain oscillatory dynamics [15–19]. These studies distinguish two alternating states in attentional fluctuation: exploitation versus exploration [16,20]. Exploitation refers to attending to the cued location and it has been associated with top-down signals, whereas exploration consists of attending to the uncued location and is considered a bottom-up process. Overall, evidence mainly comes from local recordings of prefrontal and parietal areas in non-human primates, although a few studies have shown fluctuations also percolate visual regions [21–22].

What is striking is the wide range of frequencies for which attentional sampling has been reported, ranging from 3 to 26 Hz, thereby encompassing theta, alpha, and beta rhythms [12,15,17,20,21]. One explanation for this variety might be that the rhythm of attentional sampling is not stationary. That is, the frequency at which the AS moves may change over time. Thus, the orienting period could be initially dominated by an exploration-exploitation alternation state that samples the visual field to avoid missing important but unexpected information in a bottom-up manner. Once the visual field has been explored, a stable exploitation state may set in where attentional resources are allocated to the location of interest. This latter state would likely be guided by a top-down signal and drastically reduce the frequency of attentional sampling.

Our study aims to shed light onto the neural dynamics of the AS by relating conventional and novel analysis approaches. Thus, we combined the analysis traditionally employed on this topic, ERPs and time-frequency (TF) analysis, together with a time-sensitive decoding approach that tracks the location of attention on a millisecond-by-millisecond basis in parieto-occipital electrodes during an informative cueing paradigm. Moreover, we tested these dynamics in two different datasets, one with peripheral cues and another one with central-symbolic cues, to control for purely sensory-driven processes. Multivariate pattern analysis (MVPA) allowed to decode whether attention at any given timepoint is deployed to the cued location, which is either the left or the right visual field (LVF or RVF), using high-density electroencephalographic (EEG) recordings in humans.

First, we obtained temporal generalization matrices (TGMs), by training and testing a classifier on every combination of timepoints for both cued hemifields. Such TGMs are ideal to reveal neural code generalizations, that is, neural generators that are reactivated across time [23]. Fig 1 illustrates three hypothetical TGMs according to the scenarios mentioned above, AS as: (i) a constant process (Fig 1A), (ii) a stationary dynamic process (Fig 1B) or (iii) a quasi-rhythmic dynamic process (Fig 1C). The first scenario would give rise to a sustained pattern, in which high accuracy spreads off the diagonal over time (Fig 1A), indicating that one neural pattern tracks the location of attention constantly. The second scenario would lead to an oscillatory pattern reflecting rhythmic sampling between locations at a stationary

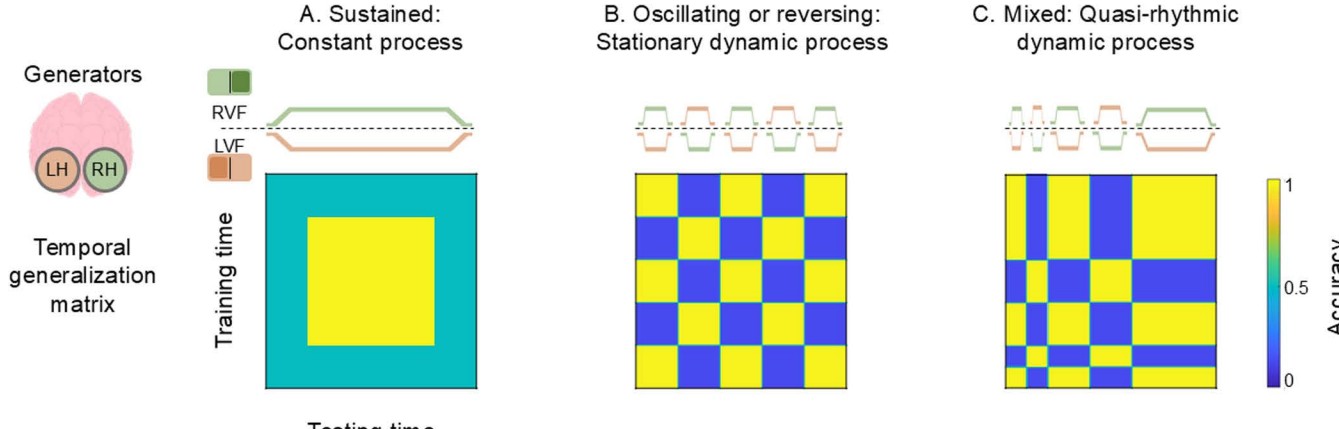

**Fig 1. Schematics of different hypothetical scenarios of attentional sampling.** The upper panel shows the tracking of the visual field (right, RVF and left, LVF) by the neural generators from the right and left hemispheres (RH and LH) across time. The time axis starts with cue onset and ends with stimulus onset. The lower panel depicts the temporal generalization matrix corresponding to each possible scenario. A. Sustained pattern: attentional spotlight (AS) as a constant process, i.e., constant fixation. B. Oscillating pattern: AS as a stationary dynamic process, i.e., sampling information from LVF and RVF at a fixed rhythm. C. Mixed pattern: AS as a quasi-rhythmic dynamic process, i.e., decreasing sampling rate until stabilization.

rate. As shown in Fig 1B, the resulting TGM would be a yellow-blue checkerboard where the classification criterion at a given training time goes from above chance classification at one testing time to below chance classification at another, which results from the shift of the AS from one visual field to the other one. However, diagonal performance would remain above chance. Under the third scenario, sampling would evolve from shifting between hemifields at a decreasing rate to settling onto the cued hemifield. In terms of the TGM, this would mean a transition from a yellow-blue checkerboard to a larger yellow blob (Fig 1C).

Eventually, we implemented the opposite approach to identify the TGM pattern to expect under the modulations of the electrophysiological attentional correlates, that is, ERPs and alpha activity. Thus, we simulated two datasets with the same experimental conditions, attending to the left or right visual field (LVF or RVF), mimicking the modulation expected on the electrophysiological attentional correlates: contralateral P1/N1 complex and contralateral decrease and ipsilateral increase of alpha, both in occipital channels.

## 2. Materials and methods

### 2.1. Participants

Dataset 1 consisted of twenty-seven healthy students from the Autónoma University of Madrid with a normal or corrected-to-normal vision and was collected between April 19 and May 25, 2018. Data from 3 participants were excluded due to technical problems during EEG recordings, yielding a final sample of 24 participants (20 females; 20 right-handed; mean ± SD age: 19.5 ± 1.9). Students were compensated with course credits for their participation. They all provided written informed consent before the experiment. The Autónoma University of Madrid's Ethics Committee approved the study (Project ID: CEI-85-1581), which was conducted in compliance with the Declaration of Helsinki.

Dataset 2 was obtained from [24] with the approbal of Dr. Poch, where thirty-six participants took part (20 females; 36 right-handed; mean ± SD age: 20.9 ± 2.7). Due to missing trigger information in the EEG files, 11 participants were excluded, resulting in a final sample of 25 participants. This dataset was accessed April 12, 2023 and the authors did not have access to information which could identify individual participants.

## 2.2. Stimulus material, experimental procedure and EEG recording for dataset 1 (peripheral cue)

**2.2.1. Stimulus material.** Stimuli consisted of Gabor patches generated offline using Matlab (R2018a, The MathWorks). Vertically oriented gratings were defined as the target stimuli, with a 10% probability of appearance, while standard stimuli were angled at ± 45°.

Stimuli were presented in 24 locations in the visual field (see Fig. 2A), which was divided according to different polar angles (in 45° steps) and eccentricities (three concentric rings of 2.6°, 9.8° and 22.2° radius), based on [25]. Stimuli were scaled to the respective 24 locations to compensate for cortical magnification [26–28]. The polar angles and eccentricities stimulated in each hemifield were related to the study of the Gabor-induced brain response, which is beyond the aim of this work. Instead, we here focus on the brain response to the attentional cue.

**2.2.2. Experimental procedure.** The study was conducted inside a dimly lit, sound-attenuated and electromagnetically shielded room. The experimental task was carried out using PsychToolbox [29–30] and presented on a 55-inch monitor, allowing the stimulation of peripheral regions of the visual field. Participants were comfortably seated 80 cm away from the monitor.

Subjects performed a spatially cued discrimination task while maintaining their gaze on a central fixation cross. The task was divided into blocks, the structure of which is depicted in Fig 2B. At the beginning of each block, the border of one sector was illuminated for 200 ms, cueing the location to be attended throughout the block (a 100% valid cue). After an interval of 800 ms, a sequence of 20 Gabors pairs was presented for 150 ms each, with a variable inter trial interval ranging from 400 to 600 ms. One of the Gabors in a pair appeared at the cued location while the other was presented at a random uncued location, with a jitter of ± 100 ms relative to the previous one. The presentation of these Gabor sequences was not part of the purpose of this study, which focused exclusively on the orienting period from cue to Gabor onset (1 s). Every sector of the visual field was cued in five blocks, leading to 60 blocks for each hemifield. Stimulation was presented over a gray background.

Participants were instructed to pay covert attention to the cued location while ignoring other locations. They were asked to report target appearance, by pressing a key as quickly and

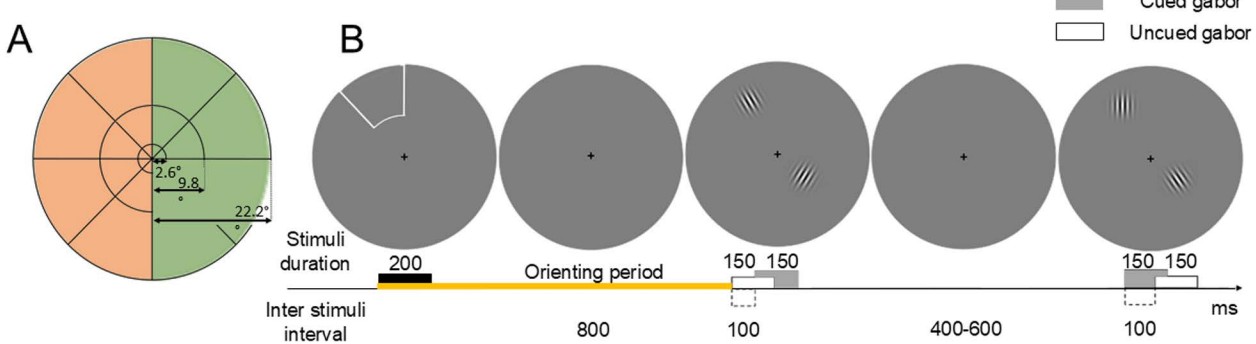

**Fig 2. Experimental stimuli and task for dataset 1 (peripheral cue).** A. Stimulated locations. The visual field was divided into 24 scaled sectors of different polar angles and eccentricities. Orange and green shades show the experimental conditions: left and right visual field (LVF and RVF, respectively). B. Example of one experimental block. Each block started with an exogenous valid cue. Participants were instructed to covertly pay attention to the cued location and ignore the other locations. After 1s orienting period, highlighted in yellow, a sequence of 20 pairs of Gabors were presented at both cued and uncued locations. Participants were asked to report cued vertical Gabors. Note that the size of stimuli is not scaled here for visualization purposes.

accurately as possible. To avoid fatigue, the experiment was administered in four runs, with self-paced breaks between them. In addition, participants were allowed short breaks between blocks, with the screen turned gray for 3 seconds. They were encouraged to restrict blinks to these periods, remaining still and relaxed. Before starting the experiment, subjects completed two blocks of practice sessions to familiarize themselves with the task. The experiment lasted approximately 40 minutes.

**2.2.3.EEG recording.** The EEG signal was recorded using a BioSemi bioactive electrode cap with 128 electrodes. Online EEG signal was referenced to one external electrode at the nose-tip and sampled at 1024 Hz. We also monitored blinks, vertical and horizontal eye movements by an electrooculogram (EOG) recorded bipolarly from the top and the bottom of the right eye and ~1 cm lateral to the outer canthi of both eyes. Offset of the active electrodes were kept below 25–30 millivolts.

## 2.3.Stimulus material, experimental procedure and EEG recording for dataset 2 (central-symbolic cue)

Dataset 2 was obtained from [24], where all the details about the task and the data collection can be found (see Fig. 3). What is revelant for the current study is that a set of four rectangles were presented at four spatial locations (top left and right and bottom left and right) after a fixation cross which was shown for 1 s. The rectangles had different colours and orientations which participants were asked to remember. A symbolic cue appeared 1 s later for 200 ms. The colour of the symbolic cue indicated which of the previously shown items will be tested. Attention was expected to be guided by the colour of this retro-cue, and thus shifted to the location of the colour-matching rectangle. After a delay of 1 s, the target rectangle was presented again for 1.5 s (100% valid cue) and participants were asked to report whether its orientation was the same as originally shown or not. A total of 160 trials were presented (80 retro-cues pointing to the LVF and 80 to the RVF).

## 2.4. Data analysis

After acquisition, data preprocessing and analysis was performed in MATLAB using FieldTrip [31], MVPA-light [32], and in-house written MATLAB code. Overall, the EEG signal analysis aimed to characterize the temporal dynamics of attentional sampling during the orienting

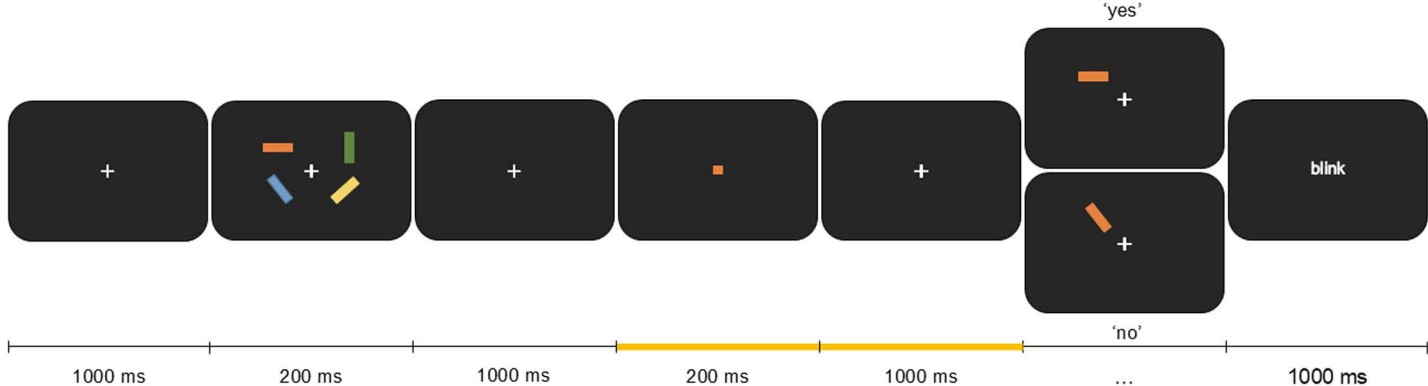

**Fig 3. Experimental stimuli and task for dataset 2 (central-symbolic cue).** Each trial started with a set of four couloured rectangles. A cue appeared later, whose colour matched one of the previous rectangles, thus indicating the location to covertly attend to. After a 1.2 s orienting period, highlighted in yellow, the target rectangle reappeared in the same location. Participants were asked to report whether its orientation was the same as previously shown or not. Fig adapted from [24].

period, i.e., in the 1 and 1.2 s period from cue to stimulus onset for dataset 1 and dataset 2 respectively. For this purpose, we first preprocessed the data by removing artifactual activity. Then, we used MVPA to decode the neural footprint of the AS over the time course of the orienting period. This classifier was trained and tested on every timepoint of posterior EEG channels which resulted in one temporal generalization matrix (TGM) for each participant. To identify the rhythm of the attentional sampling, we performed time-frequency analyses on these matrices. Then, we conducted conventional event-related potentials (ERPs) and time-frequency (TF) analysis to investigate the involvement of such neural activity in the classifier performance. Ultimately, we followed the opposite approach: we generated two datasets with the same experimental conditions as our empirical data (LVF and RVF), where underlying activity mimicked the modulations of attentional correlates, that is, ERPs and alpha. We then obtained their TGMs to reveal the pattern to expect under those electrophysiological conditions and asses the alignment between the empirical and simulated data. These proceduree are described in more detail in the following sections and is common for both datasets.

**2.4.1. Preprocessing.** The continuous EEG signal was segmented into 4500 ms epochs, starting 2000 ms prior to cue onset. The use of these long epochs allowed us to avoid edge artifacts in later time-frequency analyses. Data were low-pass filtered below 30 Hz, baseline corrected (−200 to 0 ms) and re-referenced to the common average. We did not high-pass filter the data to avoid decoding artefacts in the TGM [33]. Finally, time delays between expected and actual presentation of visual stimuli on the screen were corrected by measuring stimulus onset time with a photodiode.

Artifact rejection was carried out in a three-step procedure. First, we eliminated trials with eye movements by visual inspection, as they may indicate a failure to maintain covert attention. On average, 26.3 out of 120 trials per participant were removed (min = 5, max = 63, SD = 15.7) for dataset 1 and 15.4 out of 160 trials per participant (min = 0, max = 64, SD = 15.2) for dataset 2. Secondly, we used Independent Component Analysis (ICA) to remove remaining artifacts, related to eye-blinks or muscular activity. Finally, bad channels were interpolated using data of adjacent electrodes (dataset 1: mean ± SD of interpolated electrodes: 1.2 ± 2.7; dataset 2: mean ± SD of interpolated electrodes: 1 ± 0.7).

After artifact rejection, data were down-sampled to 128 Hz to reduce computational time and split into two experimental conditions, collapsing eccentricities and polar angles: left or right visual field (LVF or RVF). Note that initially in dataset 1 participants were asked to deploy their attention to 24 spatial locations (4 foveal, perifoveal and peripheral locations with 4 different polar angles per hemifield), while in dataset 2, attention was shifted to 4 spatial locations (top left and right and bottom left and right). Finally, we selected 18 posterior channels (P5/P6, P7/P8, P9/P10, PO7/PO8, PO9/PO10, PO11/PO12, I1/I2, POI1/POI2 and O1/O2) where attentional effects have traditionally been described [4,8,9].

**2.4.2. Multivariate pattern analysis: Temporal generalization analysis.** To decode the neural dynamics of attention in the visual field, we used the temporal generalization method [23]. Before classification, we kept trial numbers between LVF and RVF condition constant by random selection, which resulted in an average of 84.5 ± 19.2 (mean ± SD) epochs per participant in dataset 1 and 86.6 ± 21.4 (mean ± SD) in dataset 2. Then, a linear discriminant analysis (LDA, [34]) was used as a classifier. This classifier was trained and tested for each participant on every timepoint by using the z-scored amplitude from the selected posterior EEG channels as features. This resulted in a temporal generalization matrix (TGM) with one value for every combination of training and testing times. A TGM not only indicates when brain activity differs between experimental conditions, but also if a neural pattern present at a given timepoint is reactivated at any other time, shedding light on the temporal organization of information processing. To avoid overfitting, this procedure was applied following a

cross-validation scheme [35] where the data were randomly split into five folds. Four of these folds were first used as the training set to find the decision boundary that best separated both classes. Then, that boundary was applied to the fifth fold, i.e., the testing set, to predict which class the neural pattern belonged to. The process was repeated until each fold was used as the testing set once. In addition, to avoid obtaining a biased result due to the random folding and to ensure a stable frequency measurement in later steps, the analysis was repeated 15 times with new random folds and then averaged across those 15 runs.

The classification metric was accuracy, which is the fraction of trials correctly predicted by the classifier (being 50% considered chance level). We used a non-parametric two-level permutation approach to test when the classifier performance was statistically significant. To this end, we first created 50 permuted TGMs per participant by randomly shuffling the classification labels (LVF and RVF). We z-scored both real and shuffled TGMs for each participant according to the mean and standard deviation of their shuffled TGMs and used these z-scored TGMs in all subsequent steps as well as in the further frequency analysis. Then, we generated a random distribution for each testing time by training time combination over 10,000 iterations. In each repetition, we randomly drew one of the 50 z-scored shuffled TGMs per participant and computed a grand-average TGM (i.e., mean of the 24/25 randomly selected TGMs, accordingly to each dataset) under the null hypothesis. The resulting distribution of the means was used to derive the empirical p-values of the grand-average real TGM. Thus, p-values below/above the 2.5th/97.5th percentiles in each point of the TGM were considered statistically significant. P-values were previously corrected for multiple comparisons by False Discovery Rate [36].

This statistical approach aimed at detecting dynamic patterns in the data, being highly stringent in terms of multiple comparison corrections. Thus, constant processes over time could result in false negatives. Therefore, when this procedure led to non-significant results, we used cluster correction in *MVPA Light*, with 1,000 permutations to test for significant clusters.

**2.4.3.Multivariate pattern analysis: Time-frequency analysis of the TGM.** We then conducted a time-frequency analysis over the TGMs to identify the rate at which attentional sampling occurs throughout time. For this purpose, each participant´s TGM was decomposed using a Hanning-tapered sliding window Fourier transform. Oscillatory power was estimated from 0 to 1 s (dataset 1) or 0 to 1.2 s (dataset 2) for each time sample (steps of 7.8 ms) at 1 Hz steps between 2 and 30 Hz. The width of the Hanning window was adjusted to 5 cycles per frequency. Note that we had segmented the data into 4.5 s epochs to avoid edge artifacts and that the classifier was trained and tested on the EEG data, so that both phase and power were taken into account. In addition, we had z-scored the individual TGMs according to each participant's shuffled mean and standard deviation.

This analysis was applied independently to each temporal series of the TGM (i.e., to each row and column; see Fig 4, left), leading to two three-dimensional matrices: training time x testing time x frequency. On the one hand, the rows matrix stored the power decomposition of each frequency in the same training time and along testing time (Fig 4, top-middle). On the other hand, the columns matrix stored the opposite result, i.e., power decomposition in the same testing time along the trainingtime (Fig 4, bottom-middle). Subsequently, both matrices were averaged resulting in a single three-dimensional power matrix of dimensions training time x testing time x frequency (Fig 4, right). In this way, we obtained a single value for each voxel where the power of the rows and columns was equally represented.

To assess the results of the time-frequency analysis statistically, we used the same non-parametric two-level permutation approach as described above. Time-frequency representations of the 50 z-scored permuted TGMs for each participant were computed as explained

above. Over 10,000 iterations, one shuffled power TGM per participant was randomly selected in each repetition to generate a grand-average TGM. This led to an empirical distribution where FDR corrected p-values below/above the 2.5/97.5th percentiles in each point and frequency of the TGM were considered statistically significant. Prior to the multiple comparison correction, we applied the significant mask for the classifier performance, so we only took into account the power values at significant time points. Finally, we selected the significant dominant frequency peak at each significant timepoint, reducing the matrix to two dimensions. Thus, the resulting masked TGM in Fig 6C shows the significant dominant frequency peaks at timepoints where classifier performance was above chance level.

In dataset 1, this statistical analysis revealed two patterns along the TGM main diagonal (see Classification results section below). On the one hand, two early blobs with a significant dominant frequency at 10 Hz. On the other hand, a large later blob whose dominant frequencies corresponded to 2 and 3 Hz and, to a lesser extent, faster frequencies (from 11 to 14 Hz). Then, we tested significant differences in the sampling rhythms (from 2 to 30 Hz) between both periods (0–200 ms and 450–650 ms) by means of two-factor repeated-measures analysis of variance (ANOVA).

**2.4.4.Conventional analysis: Event-related potentials and time-frequency analysis.** Event-related potentials (ERPs) and Time-Frequency (TF) analysis were applied to shed light onto the neural activity underlying the classifier's performance. Thus, these analyses were computed using the same artifact-free trials randomly selected to calculate the TGMs. Both ERPs and TF analyses were applied independently for each condition (LVF and RVF) and participant. ERPs were baseline corrected from –200 to 0 ms. Regarding the TF analysis, alpha-band power (8–12 Hz) was calculated using the same approach and parameters that were applied to the TGMs: a Hanning-tapered sliding window Fourier transform, adjusted to

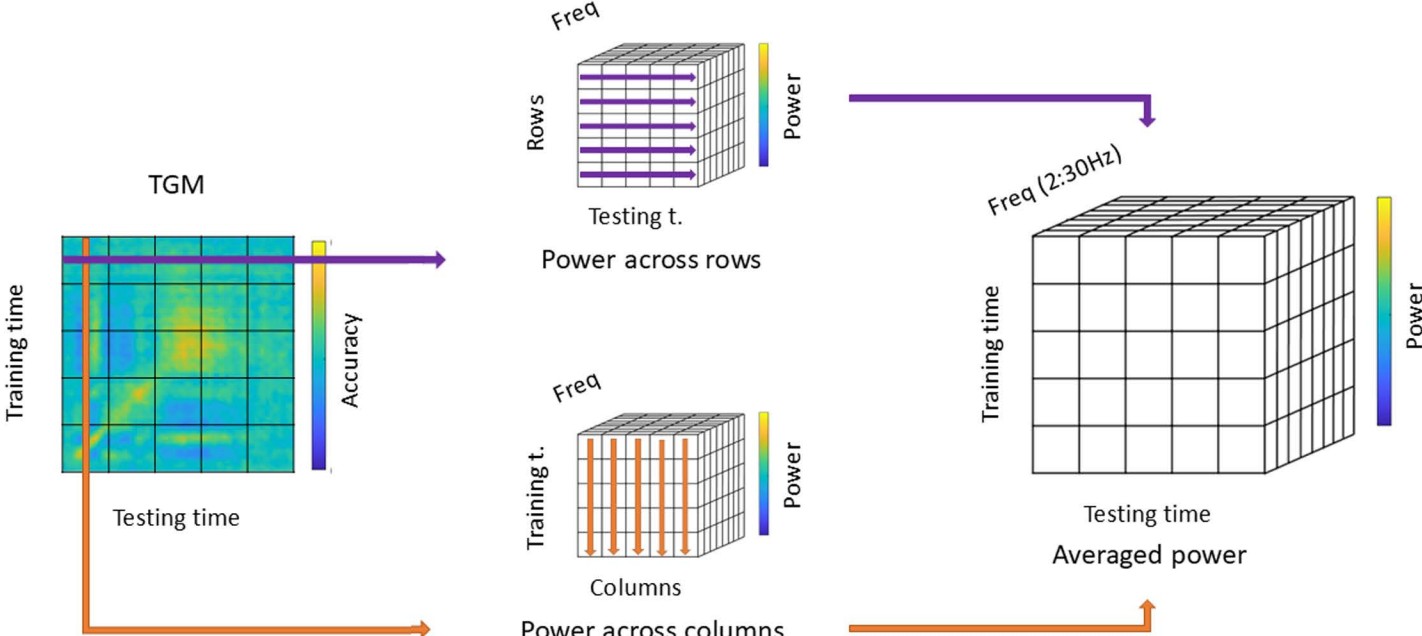

**Fig 4. Time-frequency analysis of the TGM.** We applied a time-frequency analysis independently over each row and column of the TGM (left). This resulted in two power matrices, one for the testing time evolution (top-middle) and another one for the training time evolution (bottom-middle). Both matrices were averaged to yield one composite matrix that quantifies the distribution of classifier frequencies in the TGM (right).

5 cycles per frequency, from 0 to 1 s (dataset 1) and 0 to 1.2 s (dataset 2) for each time sample at 1 Hz steps.

**2.4.5.Conventional analysis: Lateralization index.** Previous literature determined attentional effects by lateralized activity. Therefore, we employed the event-related lateralization formula to calculate a lateralization index (LI) for ERPs and alpha-band activity [37–39]. This method cancels out the neural activity not related to the AS, so it is considered as highly specific for changes in spatial attention.

$$LI = \frac{\left(left\ cue\ \left(V/P_{contralateral\ electrodes} - V/P_{ipsilateral\ electrodes}\right) + \left(right\ cue\ \left(V/P_{contralateral\ electrodes} - V/P_{ipsilateral\ electrodes}\right)\right.}{2}$$

In this calculation, the time series voltage $V$ or power $P$ after each cue onset at the ipsilateral electrodes to the cued hemifield is subtracted from the analogous voltage or power at contralateral electrodes. Subsequently, the results of both cues are avaged (LVF and RVF). ERPs contralateralization would result in values with the same sign as the component polarity, while alpha contralateralization would yield a negative index

**2.4.6.Lateralization index—Classifier accuracy correlation.** Finally, we tried to relate ERPs and alpha-band activity to the classifier performance. First, we used cross-correlations, not only to determine how well these measures match each other and at what point, but also to reveal possible periodicities previously described in our hypothetical scenarios of attentional sampling (see Fig 1). Thus, we correlated the grand average of ERPs and alpha-band LI with the average TGM diagonal accuracy in the analized temporal window formerly analized, sliding the data every time sample.

The cross-correlation results revealed that the best match between the time series was around 0ms-lag (see results section), so we subsenquently computed the correlation

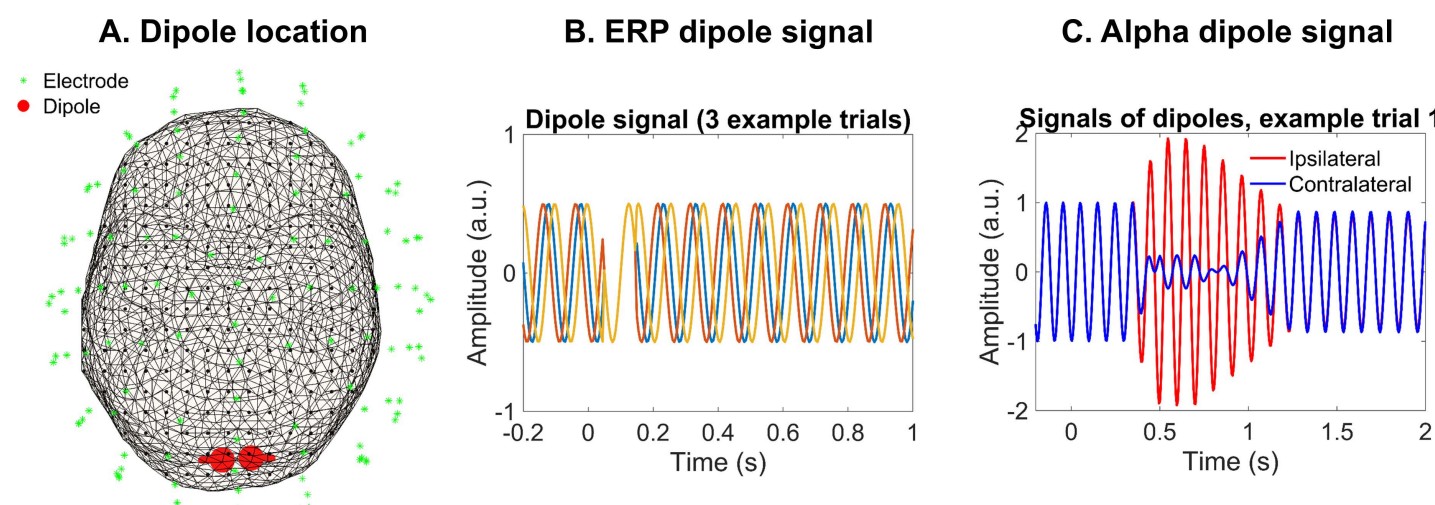

**Fig 5. Simulated data.** A. Dipole location. This image shows both the location of the two dipoles used for the simulations (left and right visual cortex) and the electrodes on which the dipole signal was projected. B. ERP simulated data. The signal of the dipole that was used to generate each experimental condition is shown over three trials. It starts with a random phase, which is interrupted by a stimulus-induced phase reset. C. Alpha simulated data. This panel depicts one trial of the two dipole signals used to generate each cue condition (red and blue line). Both dipoles are in phase and, after the onset of the presumed cue at 0 ms, the contralateral one shows a power decrease (blue line) while the ipsilateral one increases in power (red line).

coefficient between TGM diagonal accuracy and ERPs/alpha-band LI, timepoint by timepoint along the orienting period over participants. To statistically test this correlation, we applied a permutation approach over 10,000 iterations, where we shuffled the ipsilateral and contralateral channels before calculating the LI. A cluster-based multiple comparison correction was used.

**2.4.7. Data simulation.** Finally, we simulated two datasets with the same conditions of our empirical data, orienting attention to the left visual field (LVF) or to the right (RVF), to provide evidence of the relationship between the TGM pattern and the attentional correlates (i.e., occipital ERPs and alpha activity). The underlying brain activity between conditions systematically differed, following the attentional modulations in ERPs and alpha oscillations (Fig 5). This would grant us complete control over the ground truth brain patterns to generate TGMs which can then be compared with the empirical data.

To simulate the data with ERPs, we placed two homologous dipoles oscillating at 10 Hz in the visual cortex (in the left hemisphere to generate the RVF condition: x = −3, y = −93, z = 12, in the right hemisphere for the LVF condition: x = 11, y = −92, z = 12, see Fig 5A). Dipole orientation was mirrored for the two sources such that each diploe was oriented in opposing lateral directions (i.e., [−1 0 0] and [1 0 0] for left and right source, respectively). The dipole signal had a random phase across 60 trials during 4.6 s (time vector from −2 to 2.6 s, sampling rate 200 Hz). Then, we simulated a stimulus-induced phase reset: the phase of every trial shift to 0° at 70 ms, which was expected to result in a positive and negative peak between 100 and 150 ms after averaging, resembling the complex P1/N1. Eventually, a phase desynchronization was induced at 170 ms to prevent the ERP from continuing indefinitely (Fig 5B). We also injected the data with some variability to emulate natural brain dynamics (such as pink noise amplitude and frequency drifts). Using this model, the scalp distribution was generated at 128 channels (same as our empirical data). We used the 'ft_dipolesimulation' function from Fieldtrip [31]. We validated this simulated dataset by computing the average for each condition (LVF and RVF) and plotting the ipsilateral and contralateral ERPs of the same posterior electrodes used for the empirical data analysis (that is, P5/P6, P7/P8, P9/P10, PO7/PO8, PO9/PO10, PO11/PO12, I1/I2, POI1/POI2 and O1/O2). Lastly, we followed the same procedure described in section 2.4.2 to generate its TGM from these posterior channels.

The second simulated dataset was intended to mimic a contralateral decrease and ipsilateral increase in occipital alpha power in relation to the attended hemifield after a cue onset (Fig 5C). Thus, two oscillating dipoles were placed on the left and right hemifield with a frequency of 10 Hz and a random phase across trials. In each experimental condition (LVF and RVF), the contralateral dipole signal initiated a power decrease at 0.35 s, peaking at 0.5–0.7 ms and reaching previous values at 1.25 s, meanwhile the ipsilateral dipole had a mirrored increase. All other parameters remained the same as in the ERPs simulation. The validation of this dataset consisted in the estimation of the oscillatory power using a Hanning-tapered sliding window Fourier transform. The data was decomposed in 10 ms steps, obtaining frequencies from 4 to 30 Hz in 1 Hz steps and with the Hanning window width adjusted to 5 cycles per frequency. Then, we plotted the posterior contralateral and ipsilateral alpha (10 Hz) for each condition. Eventually, we computed the TGM, using the posterior channels as features (see 2.4.2 section).

The last step was to determine which of the two simulated TGM aligns most closely with the empirical TGM obtained for each dataset. We computed a 2-D correlation coefficient for each participant correlating their empirical TGM with the two simulated TGMs (ERP and alpha). This was done separately for each dataset, resulting in a single correlation value per participant, simulation and dataset. Correlation coefficients were normalized by applying a Fisher Z-transformation and their absolute values were statistically tested with a two-factor

ANOVA, with Simulation (ERP vs Alpha) as within subjects factor and Attentional Cue (peripheral vs central) as a between-subject factor. Greenhouse-Geisser correction was considered in case of nonsphericity and Bonferroni was used for multiple comparison correction. We estimated effect size using the partial eta-square ($\eta_p^2$) method.

**2.4.8.Code accessibility.** The data and scripts to reproduce the analysis and figures in this paper are available on the Open Science Framework platform (https://osf.io/8e9c7/).

## 3.  Results

### 3.1.  Behavioural results

In dataset 1, hit rates for trials in which participants were covertly oriented to the LVF (36.5 ± 10%, mean ± SD) or RVF (37.1 ± 10.9%, mean ± SD) did not statistically differ (t(23) = −0.41, p = 0.68). Likewise, reaction times for correct trials were not different (t(23) = −0.00, p = 0.99) for LVF (547 ± 52 ms, mean ± SD) and RVF (547 ± 56 ms, mean ± SD).

Dataset 2 also showed no significant differences between the two visual fields in hit rates (LVF, 81.5 ± 13.3%, RVF, 81.7 ± 13.2%, mean ± SD, t(24) = −0.16, p = 0.87).

### 3.2.  Classification results for dataset 1 (peripheral cues)

We employed a multivariate temporal classification approach to examine whether the AS samples information between LVF and RVF in two dataset (peripheral cue and central-symbolic cue). For this purpose, we trained and tested a classifier to distinguish between the two cued hemifields on every combination of timepoints. As illustrated in Fig 1, a constant spotlight would be reflected in a sustained pattern, a stationary rhythmic spotlight would lead to an oscillating pattern, and a quasi-stationary dynamic spotlight would lead to a combination of the above, with the frequency of the attentional fluctuation decreasing over time.

Fig 6 depicts the classification results for dataset 1. Representative individual TGMs are shown in Fig 6A which revealed a fluctuating dynamic. This pattern was also evident in the grand-average TGM (Fig 6B) which clearly shows that classifier accuracy fluctuated between 0.4 and 0.6 (Fig 5B). Furthermore, the non-parametric permutation test demonstrated that classification accuracy differed significantly from chance level at various points in the TGM (see Fig 6D). Importantly, significant results were obtained for both accuracies below and above chance level (see blue and yellow areas in Fig. 6B, respectively). Below chance level accuracies were confined to the off-diagonal areas of the TGM, with the diagonal showing only above chance level classification accuracies. The fluctuation of accuracy is particularly clear in the early time window (between 50 and 400 ms), and less present at later timepoints (after ~ 700 ms).

Interestingly, the left and bottom borders of the TGM show a vertical and horizontal structure respectively, with a stable pattern being present approximately from 400 to 700 ms on both time axes. This indicates that the early transient attentional state generalized to a later, apparently more sustained state, suggesting that the neural activity pattern that is present around 150 ms re-occurs in the late, sustained period. Thus, attention first appears to sample information rhythmically from both hemifields, before it settles steadily onto the cued location. These results suggest that the AS would have two sequential modes of operation, moving back and forth between hemifields during an early exploration-exploitation stage before settling towards the cued hemifield during a later single exploitation phase. Therefore, this dataset supports the third of the three scenarios outlined in Fig 1: attention as a quasi-rhythmic dynamic process.

To investigate the temporal evolution of AS more systematically, we obtained the time-frequency decomposition of the classifier performance. We applied this analysis to each row

and column of the TGM and determined the significant peak frequency when the classifier remained above chance level. If attention sampled information rhythmically but at a decreasing rate over time, then the dominant frequency of the TGM should drop across time. As shown in Fig 6D, this is exactly what can be observed. During the first 200 ms in both training and testing time, attention fluctuated at a rate of 10 Hz. As time goes on, the fluctuation rate decreased to 2 Hz at around 500 ms, with higher frequencies still prevailing at some points (from 11 to 14 Hz). Although these were the most dominant frequencies, it is noteworthy that almost all of the frequencies from 2 to 16 Hz can be observed here at different timepoints (see left panel on Fig 6C).

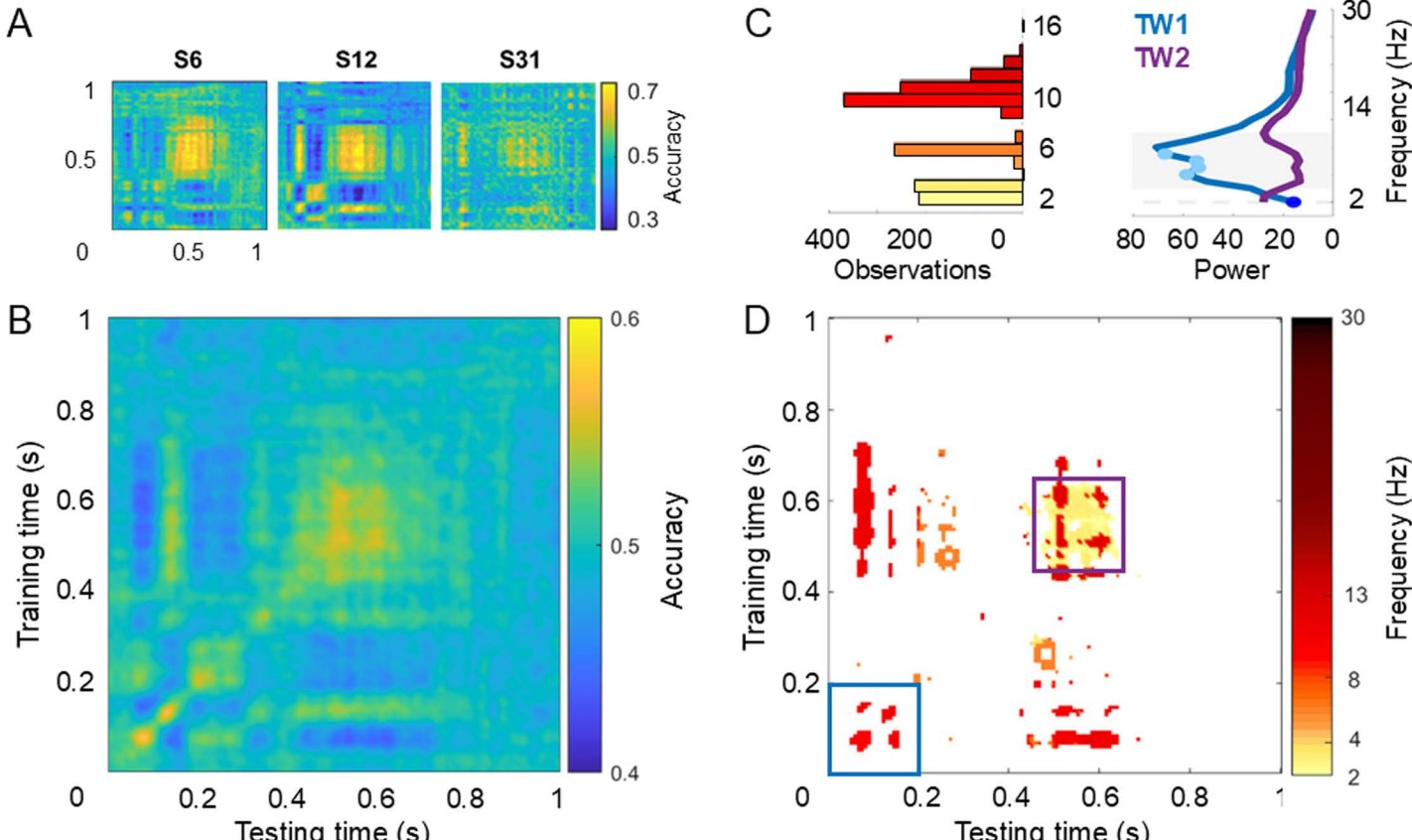

**Fig 6. Temporal classification results for dataset 1 (peripheral cue).** A. Temporal generalization matrix (TGM) for 3 individual participants. Left and right cue trials were used to train and test a classifier on every combination of time points. B. TGM averaged across all participants. C. Left panel shows the frequency distribution for significant frequency values depicted in D, represented with the same colour code. Right panel shows the means of each frequency (from 2 to 30 Hz) and time window (0–200 ms, blue line, and 450 to 650 ms, purple line). A two-factor repeated-measures ANOVA showed a significant interaction between both time windows, highlighted with a grey background while the dashed line means a marginal effect. Significant differences between frequencies in the same time window are highlighted with dots. D. Significant dominant sampling rhythm thresholded by significant classification accuracy. First, a significant mask for the classifier performance was created by means of a two-level permutation test. We generated 50 permuted TGMs per participant shuffling the experimental labels: right or left visual field. A shuffled distribution was created over 10,000 iterations, by averaging the permuted TGMs randomly selected per participant in each iteration. P-values were derived from this distribution, considering statistically significant those values below/above 2.5th/97.5th percentiles in each point. Multiple comparison correction was applied using False Discovery Rate. Thus, white areas represent timepoints with non-significant classifier performance. Second, we applied a time-frequency analysis over each participant's TGM, decomposing each time series across rows and columns independently. The power result from rows and columns was averaged leading to a single power TGM. Then, we used the same two-level permutation approach to test the significant frequencies at TGM's significant timepoints and we selected those significant frequencies with the highest power. A colour code was used to represent each significant dominant frequency. The result shows an initial sampling at ~ 10 Hz during the first 200 ms (blue square) and a later settlement around 500 ms (purple square). Note that the time axis starts with cue onset, 0 s, and ends with Gabor onset, 1 s.

Finally, a two-factor repeated-measures ANOVA performed to compare the TGM rhythms (from 2 to 30 Hz) between the early and the late time windows (0–200 ms and 450–650 ms) revealed a significant interaction ( $F_{28,644}$ = 14.6, p < 0.001, η2 = 0.4). Follow-up t-tests showed that the differences between the two time windows from 4 to 12 Hz ($t_{23}$ > 2.59, p < 0.044) was driven by higher power values in the first time window, where the dominant frequency was 10 Hz (highlighted grey background on right panel of Fig 6C). It should be noted that the differences between both time windows was marginally significant at 2 Hz, with higher power for the second time window ($t_{23}$ = -2.05, p = 0.052, highlighted dashed grey background on right panel of Fig 6C). When looking at each time window independently, differences reach statistical significance between 2 Hz and from 6 to 9 Hz for the early period (p < 0.047, dark and light blue dots on right panel of Fig 6C).

### 3.3. Classification results for dataset 2 (central-symbolic cue)

Classification results for dataset 2 are depicted in Fig 7. In this case, both individual TGMs (Fig 7A) and grand-average TGM (Fig 7B) revealed the highest accuracy values on a grid with slightly lower values from ~ 400 ms after the cue onset, spreading above and below the main diagonal.

The non-parametric two-level permutation test on the classifier performance resulted in no significant values. The severe multiple comparison correction that this approach entailed led us to apply a cluster correction, which prioritises the finding of large high accuracy groupings in the classifier performance or, in other words, constant processes. This analysis showed a cluster that basically included the grid previously described, with the highest accuracy values. Although the nature of the cluster correction would link the result to a constant attentional

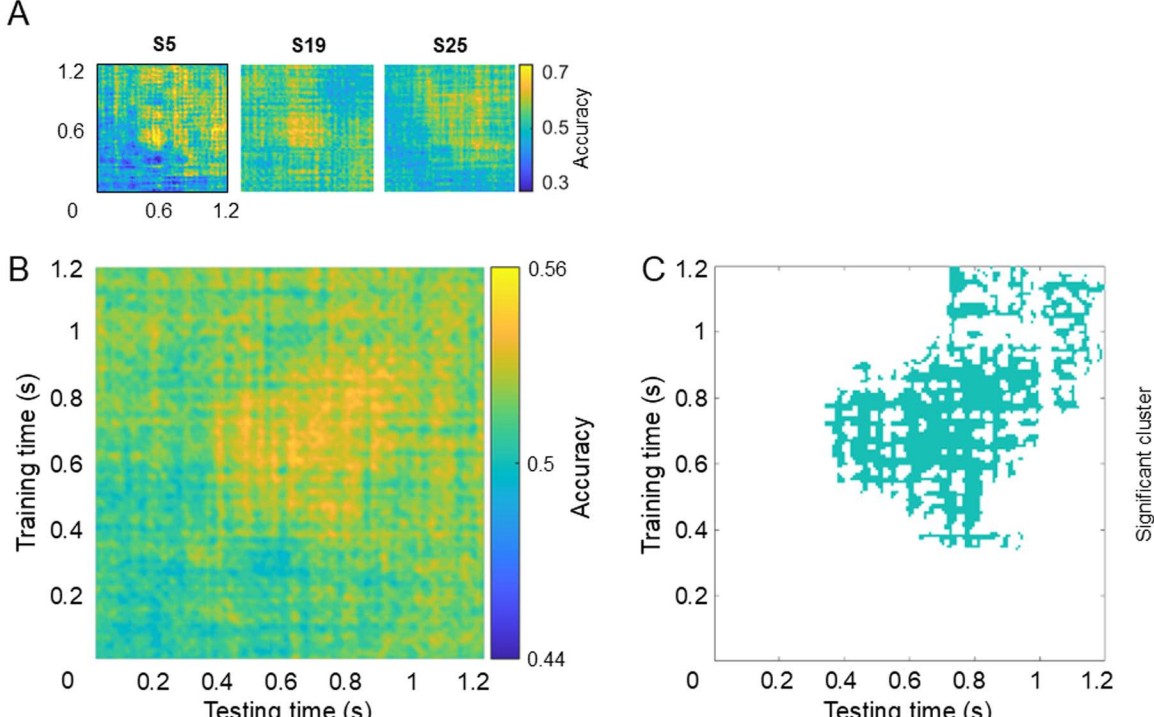

**Fig 7. Temporal classification results for dataset 2 (central-symbolic cue).** A. Temporal generalization matrix (TGM) for 3 individual participants. B. TGM averaged across all participants. C. Significant cluster for classifier accuracy.

process, we carried out the time-frequency analysis of the TGM, testing statistically only the time points with significant classification accuracy, that is, the cluster. However, the power of none of the frequencies into which the TGM accuracy was decomposed significantly differed from chance level. Therefore, this dataset supports the first of the three scenarios depicted in Fig 1: attention as a constant process.

### 3.4. ERPs and alpha results for dataset 1 (peripheral cue)

To find out whether the neural correlates of attention as measured with ERPs and alpha power are related to the classifier performance, we calculated a lateralization index (LI), independently for occipital ERPs and alpha-band power. These LIs were then cross-correlated and correlated with the accuracy values of the TGM's main diagonal independently for dataset 1 and dataset 2, shown in Figs 8 and 9 respectively, with ERPs depicted in the left panel and alpha-band power in the right panel.

Cue stimulation in dataset 1 elicited three ERP components: P1, N1 and P2 (see top image of the left panel in Fig 8). The LI, calculated by subtracting ipsilateral from contralateral activity, shows different peaks whose polarity changes coincide with the alternating yellow and blue blobs in each row of the TGM (see second image of the left panel in Fig 8). However, the cross-correlation does not have an alternating pattern matching the lag between peaks, but a sustained decreased between -125 and 215 ms-lags (third image of the left panel in Fig 8). Indeed, the correlation time point by time point between the LI and the TGM diagonal resulted in significant correlations between ~ 100 to ~ 650 ms, which includes both the early and the late TGM blobs (see bottom image of the left panel in Fig 8).

Concerning alpha-band power a pronounced power decrease for both ipsilateral and contralateral activity after the cue onset was observed in dataset 1 (see top image of the right panel in Fig 8). The LI, second image, shows a higher decrement for the contralateral alpha in the vast majority of the orienting period. Again, the cross-correlation exhibits a negative sustained pattern around 0 ms-lags (from –100 to 60 ms-lags, see third image if the right panel in Fig), but with smaller values. However, although the LI waveform appears to overlap with some TGM patterns no strong continuous correlations between alpha power LI and TGM diagonal were observed (bottom image of the right panel in Fig 8).

### 3.5. ERPs and alpha results for dataset 2 (central-symbolic cue)

The symbolic retro-cue used in dataset 2 elicited the same three ERP components, that is, P1, N1 and P2 (Fig 9, top image of the left panel). However, LI did not seem to overlap with the accuracy changes of the TGM (see second image of the left panel in Fig 9). The cross-correlation values slightly fluctuate between 0 and 0.5. However, the maximum of the temporally shifted correlations can be found with a 0 lag, which is in line with the lack of significant results in the correlation between LI and the TGM diagonal (bottom left panel in Fig 9).

Ipsilateral and contralateral alpha-band power is observed after the cue onset in the dataset 2 (Fig 9, right panel, top image). The LI, in the second raw, reflects a contralateral alpha decrease from ~ 600 ms, which coincides with the high accuracy blob of the TGM during several milliseconds. This also matches the cross-correlation sequence (third image in the right panel of the Fig 8), where the values abruptly inscrease from 0 to ~ 0.6 between -100 and 340 ms-lags, pointing to an stable attentional sampling. However, the correlation coefficient only shows isolated significant time points between alpha power LI and TGM diagonal at the beginning of the orienting period (bottom image in the right panel, Fig 9).

### 3.6.Simulation results

Two simulated datasets with the electrophysiological correlates of attention (ERPs and alpha modulation) when attending to the LVF and RVF were generated to provide evidence on the TGM pattern resulting from that ground truth. In each dataset, the experimental conditions differed on the dipole location, which were placed in the right or/and left visual cortex (see Fig 5A).

The ERP simulation consisted of a single dipole oscillating at 10 Hz in the contralateral hemisphere to the simulated attentional cue, with a phase reset around 70 ms (see Fig 5B).

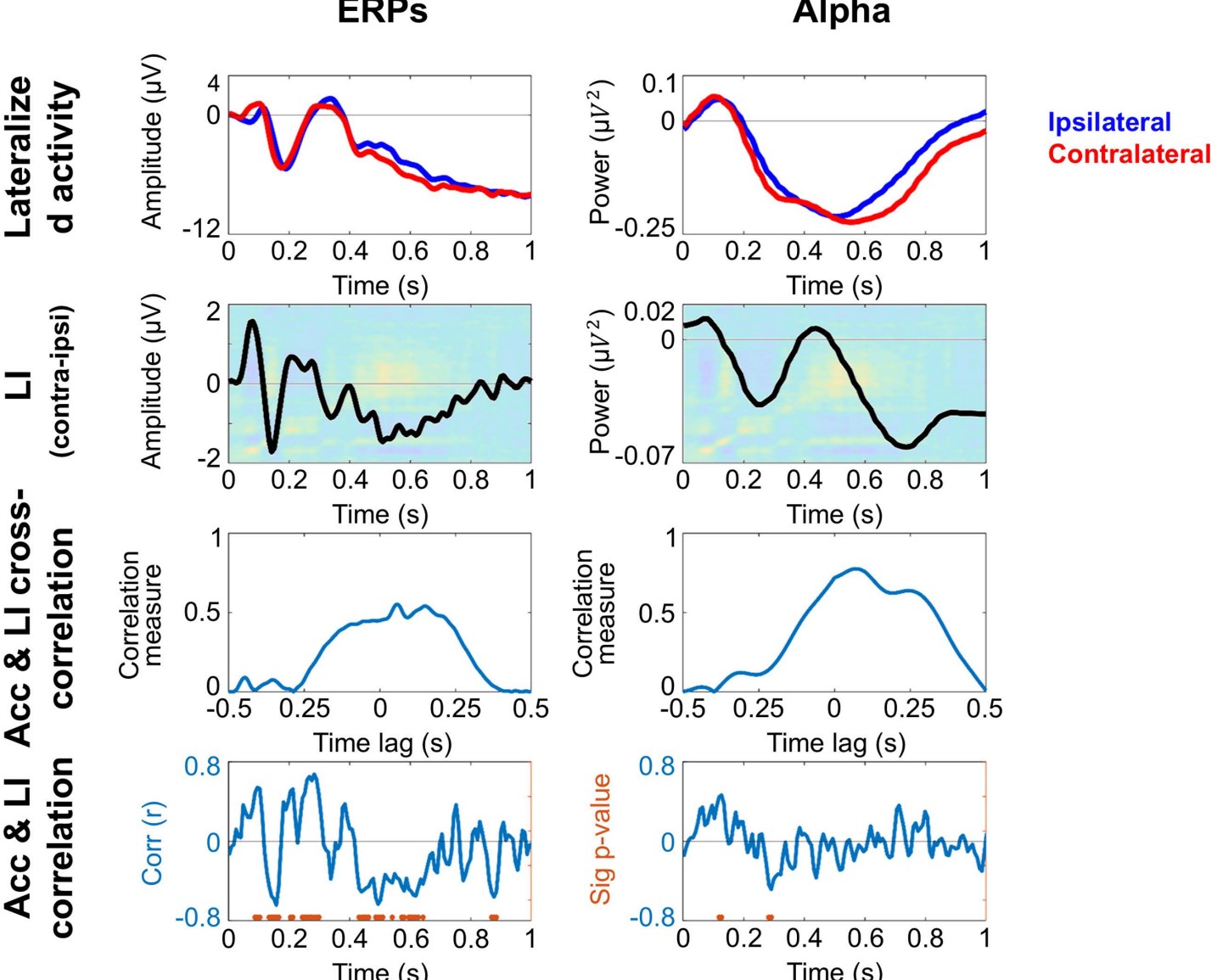

**Fig 8. Conventional univariate analysis for dataset 1 (peripheral cue).** Occipital ERPs and alpha-band activity are shown in the left and right panel respectively. The upper row depicts the lateralized activity, blue line for ipsilateral and red line for contralateral, averaged for left and right cue conditions. In the second row, the lateralization index (LI: contralateral – ipsilateral activity) is shown overlaid on the average TGM. The third row displays the cross-correlation between TGM diagonal accuracy and LI. The bottom row shows the correlation coefficient between the same measures. The orange dots highlight the time points where the correlation is significant (cluster corrected).

The dipole orientation was reversed when moving the dipole to generate the other experimental condition. As expected, when computing the average across trials and conditions (LVF and RVF) to validate the dataset, two consecutive contralateral peaks were observed in the posterior electrodes (see Fig 10A, first row). Resembling the P1/N1 complex, the first component was positive and peaked at ∼100 ms, while the second one had a negative polarity and a ∼180 ms latency. Finally, the TGM computed on posterior channels showed a reverting pattern between 70 and 170 ms, coinciding with the ERP component latency (Fig 10A, second row). The higher accuracy values were found in the main diagonal, while the lowest ones spread off-diagonal, due to the P1/N1 polarity reversion between hemispheres and conditions.

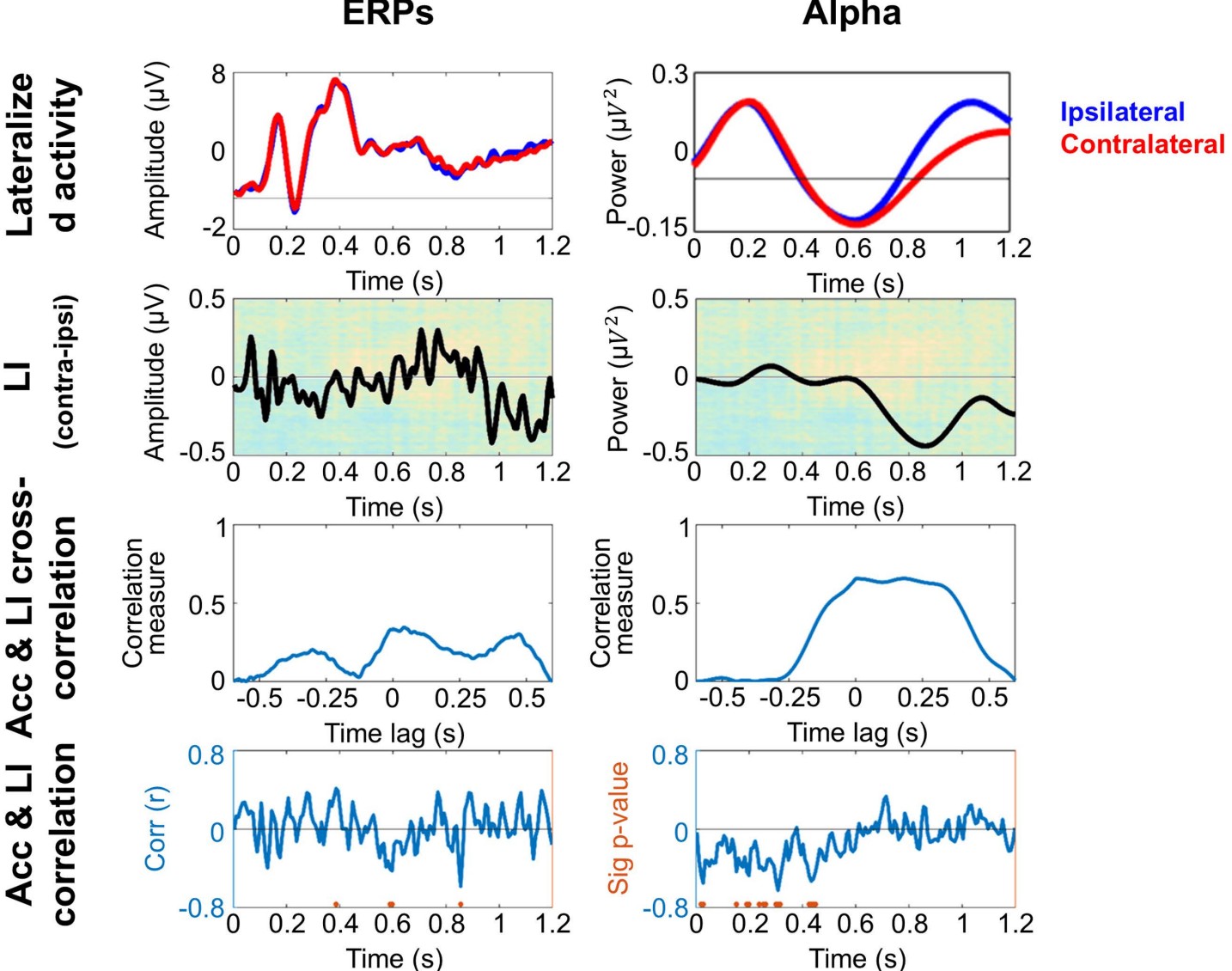

**Fig 9. Conventional univariate analysis for dataset 2 (central-symbolic cue).** Occipital ERPs and alpha-band activity are shown in the left and right panel respectively. The upper row depicts the lateralized activity, blue line for ipsilateral and red line for contralateral, averaged for left and right cue conditions. In the second row, the lateralization index (LI: contralateral – ipsilateral activity) is shown on top of the average TGM. The third row displays the cross-correlation between TGM diagonal accuracy and LI. The bottom row shows the correlation coefficient between the same measures. The orange dots highlight the time points where the correlation is significant (cluster corrected).

Regarding the alpha simulation, both experimental conditions had one 10 Hz oscillating dipole in each hemisphere. There was a power modulation from 0.35 to 1.2 s (Fig 5B), with a power increase in the ipsilateral dipole (red line) and a power decrease in the contralateral one (blue line). This modulation can be observed when decomposing the simulated signal into frequencies: posterior contralateral channels showed a decrease at 10 Hz during the same latency (Fig 10B, first row). However, this neural signature was not informative for the classifier to be able to decode the two experimental conditions, as shown by the TGM with random accuracy values (Fig 10B, second row).

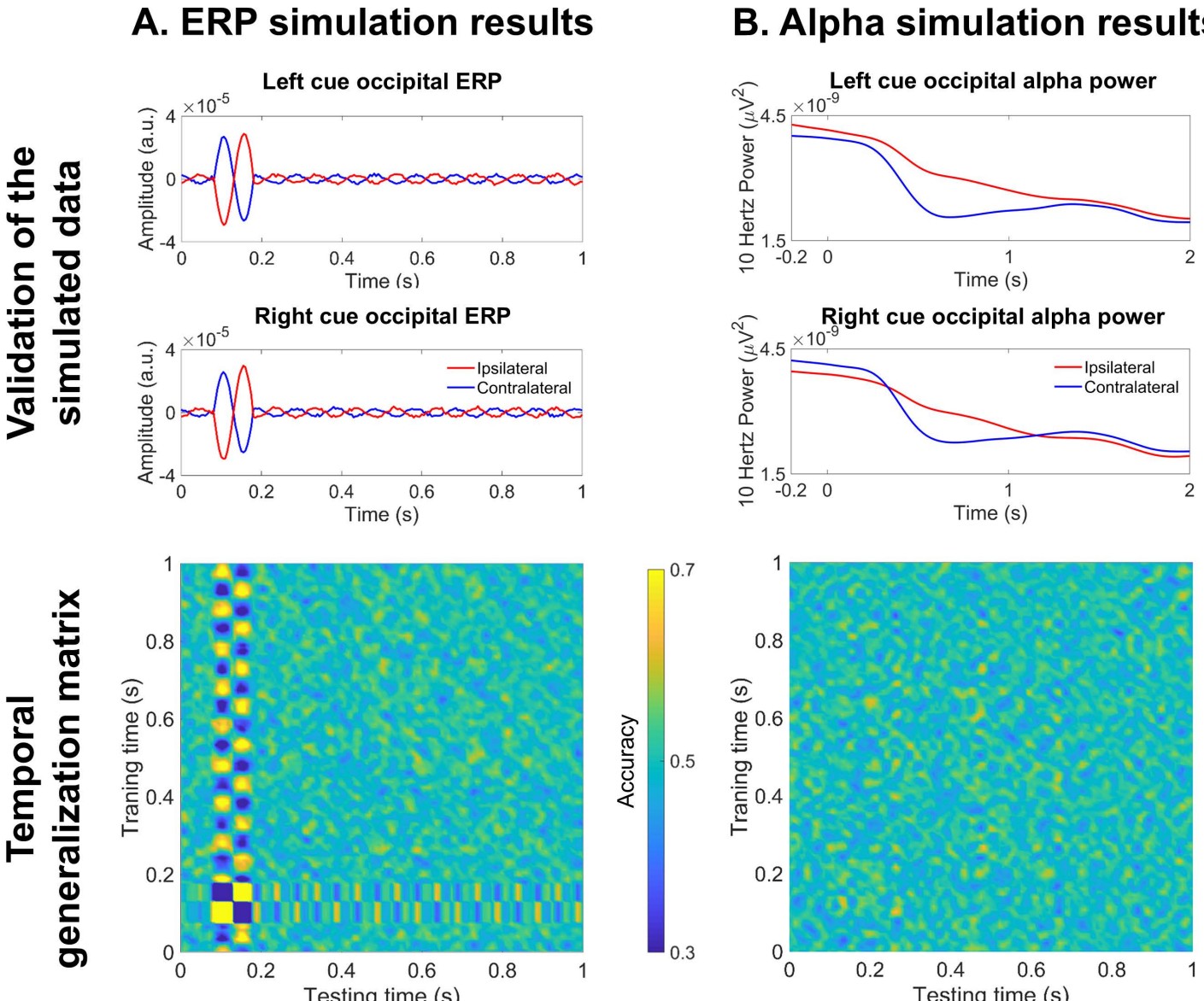

**Fig 10. Simulated data.** A. ERP simulation. The first row depicts the average of the simulated data, for each cue condition and each posterior hemisphere. A contralateral P1/N1 complex can be observed (blue line), validating this data simulation. The second row is the TGM of this simulated dataset, revealing an early reverting pattern at the P1/N1 complex latency. B. Alpha simulation. In the first row, the 10 Hz power of the simulated data has been displayed, again for each cue and posterior hemisphere. The blue line shows the expected contralateral alpha decrease. The second row depicts the TGM, where no pattern can be observed.

Eventually, we computed the correlation between the empirical and the simulated TGMs and applied a two-way ANOVA with Simulation (ERP or alpha) as a within-subjects factor and Attentional Cue (peripheral cue dataset and central-symbolic cue dataset) as a between-subject factor to assess which simulation best fits the empirical datasets. This test revealed significant main effects for Simulation ($F_1 = 8.46$, $p = .006$, $\eta_p^2 = 0.15$, $\beta-1 = .81$) and Attentional Cue ($F_1 = 5.54$, $p = .023$, $\eta_p^2 = 0.11$, $\beta-1 = .64$) and a significant interaction between both factors ($F_1 = 4.62$, $p = .037$, $\eta_p^2 = 0.09$, $\beta-1 = .56$). Specifically, when considering the best fit for the peripheral cue dataset (see Fig 11, left side), higher correlations were observed with the ERP simulation ($M = .030$, $SD = .005$) than with the alpha simulation ($M = .012$, $SD = .002$, $t_{(23)} = 2.28$, $p < .001$), but no differences were found in the central-symbolic cue dataset (see Fig 11, right side). Additionally, the coefficients obtained from correlating each dataset with the ERP simulation (see green data in Fig 11) were higher for the peripheral cue dataset ($M = .030$, $SD = .005$) than for the central-symbolic cue dataset ($M = .014$, $SD = .004$, $t_{(47)} = 2.48$, $p = .017$), while there were no differences in the alpha simulation (red data in Fig 11).

## 4. Discussion

This study aimed to shed light onto the spatiotemporal dynamics of the AS using two informative cueing paradigms, with a peripheral and a central-symbolic cue, to control for purely sensory-driven processes. In the dataset with the peripheral cue, time-frequency analysis of classifier performance revealed a non-stationary quasi-rhythmic attentional sampling process. This process started by sampling information rhythmically between cued and uncued locations at a rate of ~ 10 Hz before settling onto the cued location. The earlier process might reflect a bottom-up exploration-exploitation process, whereas the latter process might index a top-down driven exploitation state, both reflected by activity in occipital regions. The correlation analysis between conventional markers of attention and classifier results revealed that ERPs are a major contributor to these attentional dynamics. In contrast, alpha-band lateralization did not show such a strong relationship with classifier accuracy. However, the dataset with central-symbolic cues pointed to a constant attentional process, since the early rhythmic sampling found in the first dataset dissapeared, while the late static attentional exploitation

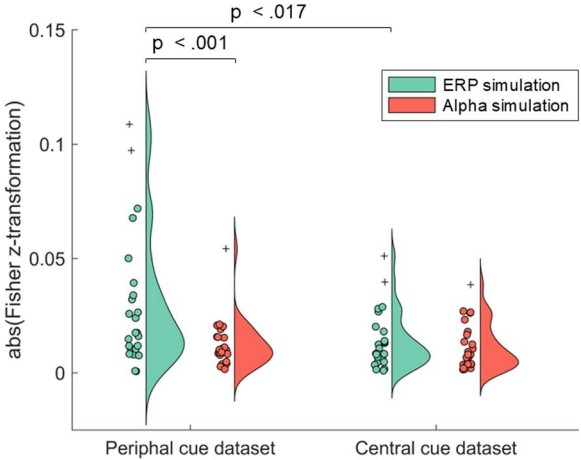

**Fig 11. Empirical data and simulation alignment.** After obtaining the 2-D correlation coefficient between each participant TGM for the peripheral and central cue datasets (left and right, respectively) and each simulated TGM (EPR - green and alpha - red), the absolute values of its Fisher z-transformation have been represented on this raincloud plot. Crosses represent outliers.

prevailed. No correlations were found between the classification results and the conventional neural correlates of attention (i.e. ERPs and alpha activity).

Previous literature has defined the AS as either a constant process in space and time [10] or a stationary oscillatory process [11–13], which samples information alternately from both hemifields at a wide range of frequencies [12,15,17,20,21]. Thus, subsequent studies assume the existence of two states in the sampling behaviour of the AS: exploitation/sampling versus exploration/shifting [16,20]. Exploitation refers to a phase where attention is being located to the cued position triggered by a top-down signal, while exploration refers to the sampling of information from uncued locations and reflects a bottom-up driven process. Our results for the peripheral cue dataset suggest that both processes are at work in an informative cueing paradigm but at different times, with an initial rhythmic fluctuation of attention at ~ 10 Hz, which subsequently, slows down to 2 Hz. More specifically, the exploration-exploitation state takes place during the first 400 ms after the cue, where attention moves quickly back and forth between hemifields at a rate of ~ 10 Hz. This could be an automatic response, i.e., that is not or only under limited voluntary control, to ensure that no important but unexpected information is being missed. Then a more stable phase settles in where attention remains focused on the cued hemifield, from 400 to 800 ms, which would be consistent with a state of exploitation where the information anticipated by the cue is being maximally extracted. We speculate that a higher-order signal is needed for the occipital AS to stop sampling from both hemifields to settle onto the cued location. For example, the prefrontal cortex is well known to be the origin of an attentional control signal [40–42]. The results are highly consistent when we move from the permutation statistical approach to describe the TGM pattern to the parametric statistical analysis to compare different time windows and frequencies. For example, alpha rhythms, suggested to be the dominant frequency in our permutation approach, are significantly higher than 2 Hz in the first time window (see Fig 6). Furthermore, frequencies in the theta and alpha band are significantly stronger in the early state than in the later one. Despite these differences in frequencies, however, due to the relatively short time window of the orienting period (1 second) the exact frequency and periodic nature of the 2 Hz effect as observed here can not be resolved. Finally, note that our method establishes the dominant significant frequency for the AS but it does not address a possible nesting of frequencies, as described in previous works [16], which could also be an explanation for the wide range of sampling frequencies reported.

The combination of a highly time-sensitive brain recording method with a powerful MVPA algorithm was key to identify fine spatiotemporal dynamics of the AS. For the current pattern to emerge it was critical to consider the TGM as a whole, instead of applying the common approach of only investigating accuracy on the diagonal. Indeed, if we had only considered the diagonal, we would have entirely missed the early alternating accuracy pattern (described in our hypotheses as the AS rhythmically going back and forth between the two hemifields). The below-chance classification accuracy on the off-diagonals TGM in dataset 1 is crucial for the results interpretation, since only there a rhythmic reversal of hemifield states after cue onset could be observed. This could be interpreted as a rhythmic attentional sampling at about 10 Hz, which would match the sampling frequencies reported in some previous studies [11,43–45]. However, other studies have reported disparate sampling frequencies ranging from 3 to 26 Hz [12,15,17,20,21]. Noteworthy, most of these frequencies were observed in the TGM computed in the peripheral cue dataset. The present approach and results might help clarifying such inconsistency, by suggesting that attentional sampling is not a stationary phenomenon but instead changes across time.

Crucially, the classification results of the central-symbolic cue dataset did not replicate the early rhythmic attentional sampling pattern. Instead, we exclusively found a big cluster coinciding with the late exploitation state, that is, spreading from 400 ms above and below the

diagonal. This suggests that it takes some time for the system to deploy attention, which then settles permanently at the cued location. Note that the main difference between both datasets is the cue type. While the first dataset has peripheral cues, the second one uses central cues. The reason for this manipulation was to control for purely sensory-driven processes triggered by the specific location of the cue in dataset 1, which would differ between the analysed experimental conditions: LVF and RVF. Hence, the use of a paradigm with central-symbolic cues, which would alleviate the visual differences between cues. Therefore, the discrepancy in the early classification results found in the two datasets could be attributed to the lateralized sensory activity elicited by the peripheral cues in the first few hundreds miliseconds, while the late exploitation state could be considered as a more genuine attentional fingerprint. Conventional analysis carried out on these datasets might provide information in this regard. It should be noted too that the use of symbolic cues has also some disadvantages. They might have slowed down the attentional dynamics or even activate additional neural patterns, making difficult to isolate the neural fingerprint of the AS.

Previous studies have found that visual areas, frontal eye-fields and lateral intraparietal sulcus are involved in attention allocation, using local recordings in non-human primates in a hypothesis-driven manner [16,20–22]. Like these studies, we used specific EEG channels based on the occipito-parietal origin of the attentional correlates [4,8,9]. However, our recordings are from human volunteers and we employed a data-driven approach (i.e., MVPA) to analyse brain activity. This suggests that visual areas also track the actual locus of attention, pointing towards the existence of an occipital AS.

Our first analysis approach and previous discussion focus directly on neural representations during the orienting period. However, despite the advantages of this data-driven alternative, this analysis does not reveal the brain activity underlying the classifier. Thus, we tried to link the MVPA results to conventional measures of EEG activity such as ERPs and time-frequency analysis. Particularly, we were looking for a periodicity-sensitive prediction measure, so we used a cross-correlation. However, a steady decreasing pattern around 0ms-lag was systematically found, making the time point by time point correlation coefficient the most informative measure. The results of the peripheral dataset suggest that lateralized activity in the ERPs contributed to the classifier performance. Specifically, correlations arose in early time-windows coinciding with the P1/N1 components, and in a later window coinciding with a slower component. However, the P1 and N1 components' involvement is unclear, since they are not only sensitive to the physical features of the visual stimulation, but are also strongly modulated by attention [5,6,46]. The results of the central cue dataset are key to discern between these two potential explanations. Indeed, central cues triggered similar ERPs components, but these did not correlate with the classifier accuracy. This is a critical finding for the data interpretation: both the peripheral and central cue involve attentional deployment, so the disparity in the results must be triggered by their physical differences. Thus, the early exploration state which is only present after the peripheral cue onset must be driven by the characteristics of the visual stimulation (i.e., the cue location) rather than attentional dynamics. Regarding alpha power, we did not find strong evidence for its contribution towards classifier performance in any of the datasets. This is somewhat surprising given that lateralized alpha power is a prominent correlate of attention [7–9].

Importantly, the simulated data provide insight into the debate regarding the underlying brain activity that leads to specific TGM patterns. Our ERP simulated data shows that a stimulus-induced phase reset, or in other words, an ERP, simultaneously in different hemispheres can be decoded by a classifier. This is our speculation regarding the empirical peripheral cue dataset, where both the ERP modulation and the simultaneous reverting TGM pattern can be observed, with a significant correlation (Fig 8). However, there is not such a reversion

in the TGM of the central cue dataset, where we controlled for purely sensory-driven processes related to the cue onset. Thus, although the P1/N1 component is evident (Fig 8), the classifier result suggests that it must have different characteristics, as there is no successful decoding. However, the scenario in relation to alpha is simpler, whose simulation leads to the same results as both empirical data: the power modulation in alpha is not detectable by the classifier in order to distinguish both experimental conditions (see Figure 7 and 8). Indeed, our analysis of the fit between the simulated and empirical data points out that the peripheral cue aligns better with the ERP simulation compared to the alpha simulation, supporting the idea that ERP activity drives the observed pattern. Therefore, our simulations support the conclusions drawn by relating the electrophysiological correlates of attention to the classifier performance.

An additional remark concerns cue validity. Previous studies describing the AS as a stable oscillatory process have used both informative and non-informative cues [12,16,20]. By using time-sensitive decoding methods, the results of the current study support the same idea when a 100% valid informative cue is employed. However, the expectations generated by cue validity [47], likely will affect the dynamics of this attentional behaviour. It is conceivable that the single exploitation state as observed here has been triggered by the high predictive value of the cue, whereas exploration-exploitation alternation may prevail when using intermediate values of cue validity. Future studies can use the method described here to disentangle between these hypotheses to further improve our understanding of attention.

There are several limitations to this study that need to be considered. First, our evidence for the dynamic displacement of the AS relies purely on neural data, as opposed to behavioural data [11–13,48]. However, measuring such rhythmic processes using behaviour requires a special experimental setup that would interfere with the neural recordings, and has also been disputed [14]. That is, we would need trials with different time length, as opposed to those required for the TGM computation. Furthermore, this would imply increasing considerably the number of trials, in order to have enough observations at neural level (cue condition x orienting period length). Second, one might wonder whether the early exploration state found in dataset 1 is related to eye movements or even micro-sacades, meaning that the visual inspection and ICA have failed in removing them [49]. Since attentional cues have been reported to bias microsaccade directions in the following 200 ms [50–52], the classifier performance could rely on them. However, what appeared to be exploration dynamics not only show an earlier shift between hemifields, but also that they are related to contralateral sensory activity, as previously explained, while the saccadic spike potential has been related to a ipsilateral topography [53]. Finally, the study includes two experiments in order to control for purely sensory-driven processes, but the paradigms are not identical. The use of a central-symbolic cue instead of a peripheral cue was essential to have that control and two alternatives were considered: a pre-cue and a retro-cue. Both options involve working memory, as participants would have to remember the association between colour and spatial location. We opted against a pre-cue design because each colour would have been associated with a specific location (e.g., red cue – LVF and blue cue - RVF), therefore counfounding basic sensory information (i.e., colour) with attentional shifts of location, which would have left us in the same situation as with the first experiment. In contrast, the retro-cue design allowed the colour associated with each location to be varied trial by trial, therefore removing the confound between sensory information and location. Additionally, the subsequent presentation of the stored item to report its orientation would still promote the shifting of AS to that location. Therefore, the retro-cue option overcomes some of the constrains. One caveat, however, is that we assume the same lateralization processes are deployed in both paradigms of our study: the peripheral cue, which relies on attention, and the central-symbolic cue,

which involves attention and working memory. Therefore, we cannot rule out the possibility that the early rhytmic sampling process is only present in attention but not in working memory, whilst the late lateralization (non-rhytmic) is present in both. Future experiments should address this question.

## 5.Conclusion

Together, our results go methodologically one critical step beyond previous attempts to characterise the spatiotemporal dynamics of visuospatial attention, by combining classical ERP and TF analysis with rhythmic MVPA-based analyses. Importantly, our findings are supported by simulations. We provide evidence for a constant process that starts a few hundred milliseconds after the attentional cue. For attention to be effective, it needs to be controlled voluntarily to sample information from areas of interest (i.e., it needs to be under top-down control). Hence, once cue information has been processed, for which the system would need some time, a hypothetical high order signal leads to a later single exploitation stage.

## Acknowledgements

We are thankful to Ian Charest for valuable advice regarding the use of linear discriminant analysis and Claudia Poch for sharing her data.

## Author contributions

**Conceptualization:** María Melcón, Almudena Capilla, Simon Hanslmayr.

**Data curation:** María Melcón.

**Formal analysis:** María Melcón.

**Funding acquisition:** María Melcón.

**Investigation:** María Melcón, Yolanda Sánchez-Carro, Laura Barreiro-Fernández, Elisabet Alzueta.

**Methodology:** Sander van Bree, Luca D. Kolibius, Maria Wimber, Almudena Capilla.

**Resources:** Almudena Capilla.

**Supervision:** Maria Wimber, Almudena Capilla, Simon Hanslmayr.

**Visualization:** Sander van Bree, Luca D. Kolibius.

**Writing – original draft:** María Melcón.

**Writing – review & editing:** María Melcón, Sander van Bree, Yolanda Sánchez-Carro, Laura Barreiro-Fernández, Luca D. Kolibius, Elisabet Alzueta, Maria Wimber, Almudena Capilla, Simon Hanslmayr.

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
