## [Decision Letter · Decision Letter 0]

9 Dec 2024

PONE-D-24-40710Evidence for a constant occipital spotlight of attention using MVPA on EEG dataPLOS ONE

Dear Dr. Melcon Martin,

Thank you for submitting your manuscript to PLOS ONE. After careful consideration, we feel that it has merit but does not fully meet PLOS ONE’s publication criteria as it currently stands. Therefore, we invite you to submit a revised version of the manuscript that addresses the points raised during the review process.

We look forward to receiving your revised manuscript.

Kind regards,

Marie-Constance Corsi

Academic Editor

PLOS ONE

“This work was supported by FEDER/Ministerio de Ciencia, Innovación y Universidades – Agencia Estatal de Investigación, Spain (grant PGC2018-100682-B-I00) and the Research Grant FPI-UAM 2017 (UAM, Spain). We are thankful to Ian Charest for valuable advice regarding the use of linear discriminant analysis and Claudia Poch for sharing her data.”

“This work was supported by FEDER/Ministerio de Ciencia, Innovación y Universidades – Agencia Estatal de Investigación, Spain (grant PGC2018-100682-B-I00) and the Research Grant FPI-UAM 2017 - Formación de Personal Investigador de la Universidad Autónoma de Madrid (UAM, Spain).

None of the funders play any role in the study”

Reviewers' comments:

Reviewer's Responses to Questions

**Comments to the Author**

1. Is the manuscript technically sound, and do the data support the conclusions?

Reviewer #1: Yes

2. Has the statistical analysis been performed appropriately and rigorously? 

Reviewer #1: Yes

3. Have the authors made all data underlying the findings in their manuscript fully available?

Reviewer #1: Yes

4. Is the manuscript presented in an intelligible fashion and written in standard English?

Reviewer #1: Yes

5. Review Comments to the Author

Reviewer #1: Review of Evidence for constant occipital spotlight of attention using MVPA on EEG data

This study investigates the temporal dynamics of visuospatial attention, challenging the traditional view that attention remains fixed at a single location. Using EEG-based classification and conventional analyses, the authors examined how attention fluctuates over space and time during visuospatial cueing tasks with central and peripheral cues.

While this paper presents some interesting aspects, including the great practice of code sharing, I believe substantial revisions are necessary before it can be recommended for publication. The authors take an interesting approach by using multiple datasets and analytical methods to address their research question. However, many of the conclusions appear to rely on observational inference, which does not provide sufficient empirical support.

The primary issue, in my opinion, lies in the authors' effort to propose different theoretical models to interpret their results—an estimable practice —but the integration of these models with the actual findings lacks rigorous evidence. I suggest the authors explore methodologies from the field of Representational Similarity Analysis (RSA), where theoretical models are directly correlated with empirical data. This approach allows researchers to determine which model aligns most closely with the observed patterns, providing a stronger basis for their claims. Currently, the assertion that one experiment reveals a specific pattern over another relies too heavily on the subjective observation of "visually similar" results.

The analyses aimed at ruling out the effects of ERPs and Alpha on classification accuracy are another point of concern. The methods used are not optimal given the different transformations applied to these signals. I recommend using sliding window analyses and/or cross-correlation techniques to ensure a more robust interpretation of the data and to disentangle the contributions of these factors more convincingly.

I would also like to understand why the authors emphasize the distinction between conventional analyses and MVPA so strongly. As it stands, the manuscript seems to introduce more confusion about potential processes at play rather than providing clear insights into the underlying physiology. Moreover, there appears to be insufficient effort to link these findings to behavior. Identifying differences in classification accuracy is not equivalent to uncovering a physiological process; it is merely another statistical approach. Bridging these analytical results with behavioral outcomes could significantly enhance the study’s impact and relevance.

Finally, the introduction mentions a role of prefrontal and parietal regions. Why is this spatial diversity not represented in the analysis, using for instance spotlight analysis and whole-scalp decoding, to explore in more detail whether other areas might be relevant to the process the author seem to want to explore.

In summary, while I believe the approaches employed in this paper have potential, more work is needed to unify these methods and present a cohesive narrative. Strengthening the link between the proposed models, the analytical results, and behavioral evidence would greatly improve the clarity and significance of the findings.

Minor comments:

-The layout of figure 5 is unconventional (i.e. the order of A, B, C, D) and it is badly mentioned on page 18

- The intro mentions ‘recent finding’ but then cites papers from 2013, 2012, 2007, to me, only the critical review seems recent

-

6. PLOS authors have the option to publish the peer review history of their article (what does this mean? ). If published, this will include your full peer review and any attached files.

**Do you want your identity to be public for this peer review?** For information about this choice, including consent withdrawal, please see our Privacy Policy .

Reviewer #1: No

---

## [Author Response · Author response to Decision Letter 1]

4 Feb 2025

Dear Dr. Corsi,

We would like to thank you and the reviewer for your valuable time and for your helpful and constructive comments, which have been of great help in improving this work.

In the revised version of the manuscript, we have basically addressed all suggestions, including 1) simulated data to relate theoretical models to empirical data, 2) cross-correlation between the ERP lateralization index/alpha band activity and the classification accuracy to ensure a more robust interpretation of the data, and a clarification of 3) the reason for emphasizing the distinction between conventional analysis and MVPA, 4) why our methodology of analysis is unfortunately incompatible with the collection of useful behavioural data, and 5) the minor comments. The only point that has not resulted in any change in the manuscript is the one related to the whole-scalp decoding. We clarify here the explanation on the occipital channels selection, but we also provide whole-scalp decoding results, proving that it is more appropriate to keep the manuscript with the initial selection of occipital electrodes guided by our hypotheses.

A detailed explanation of how we have addressed the reviewer’s concerns has been appended in this letter, with the original reviewer’s comments in bold and our responses below. The changes made in the revised manuscript have been highlighted in blue to facilitate the review. Please, go the attached file to be able to see the different font text and the new figures.

Yours sincerely,

María Melcón and Simon Hanslmayr

(on behalf of the co-authors)

REVIEWERS' COMMENTS:

Reviewer #1: Review of Evidence for constant occipital spotlight of attention using MVPA on EEG data

This study investigates the temporal dynamics of visuospatial attention, challenging the traditional view that attention remains fixed at a single location. Using EEG-based classification and conventional analyses, the authors examined how attention fluctuates over space and time during visuospatial cueing tasks with central and peripheral cues.

While this paper presents some interesting aspects, including the great practice of code sharing, I believe substantial revisions are necessary before it can be recommended for publication. The authors take an interesting approach by using multiple datasets and analytical methods to address their research question. However, many of the conclusions appear to rely on observational inference, which does not provide sufficient empirical support.

Response: Thank you very much for both the positive comments and the feedback. We have tried to clarify the aspects pointed out in the review to improve the manuscript.

The primary issue, in my opinion, lies in the authors' effort to propose different theoretical models to interpret their results—an estimable practice —but the integration of these models with the actual findings lacks rigorous evidence. I suggest the authors explore methodologies from the field of Representational Similarity Analysis (RSA), where theoretical models are directly correlated with empirical data. This approach allows researchers to determine which model aligns most closely with the observed patterns, providing a stronger basis for their claims. Currently, the assertion that one experiment reveals a specific pattern over another relies too heavily on the subjective observation of "visually similar" results.

Response: we greatly appreciate this recommendation, as this suggestion can help us to make strong statements in the interpretation of our data. In order to integrate the results from the classification analyses with the conventional analyses, we simulated two datasets containing modulations in the electrophysiological correlates of attention, i.e., the P1/N1 complex and the alpha band. Our results show that the peripheral cue dataset fits better with the ERP than the alpha simulation and, at the same time, the ERP simulation aligns most closely with the peripheral cue dataset.

First, we have included a new paragraph in the introduction to present these analyses (page 5):

“Eventually, we implemented the opposite approach to identify the TGM pattern to expect under the modulations of the electrophysiological attentional correlates, that is, ERPs and alpha activity. Thus, we simulated two datasets with the same experimental conditions, attending to the left or right visual field (LVF or RVF), mimicking the modulation expected on the electrophysiological attentional correlates: contralateral P1/N1 complex and contralateral increase and ipsilateral increase of alpha, both in occipital channels.”

In the initial summary of the section 2.4. Data analysis, we have added a couple of sentences including the simulation analysis (page 10):

“Ultimately, we followed the opposite approach: we generated two datasets with the same experimental conditions as our empirical data (LVF and RVF), where underlying activity mimicked the modulations of attentional correlates, that is, ERPs and alpha. We then obtained their TGMs to reveal the pattern to expect under those electrophysiological conditions and asses the alignment between the empirical and simulated data.”

While the details of the data simulation are included in the following new section of the methods (pages 16, 17 and 18):

“ 2.4.7. Data simulation

Finally, we simulated two datasets with the same conditions of our empirical data, orienting attention to the left visual field (LVF) or to the right (RVF), to provide evidence of the relationship between the TGM pattern and the attentional correlates (i.e. occipital ERPs and alpha activity). The underlying brain activity between conditions systematically differed, following the attentional modulations in ERPs and alpha oscillations (Fig 5). This would grant us complete control over the ground truth brain patterns to generate TGMs which can then be compared with the empirical data.

To simulate the data with ERPs, we placed two homologous dipoles oscillating at 10 Hz in the visual cortex (in the left hemisphere to generate the RVF condition: x=-3, y=-93, z= 12, in the right hemisphere for the LVF condition: x=11, y=-92, z=12, see Fig 5A). Dipole orientation was mirrored for the two sources such that each diploe was oriented in opposing lateral directions (i.e. [-1 0 0] and [1 0 0] for left and right source, respectively). The dipole signal had a random phase across 60 trials during 4.6 s (time vector from -2 to 2.6 s, sampling rate 200 Hz). Then, we simulated a stimulus-induced phase reset: the phase of every trial shift to 0º at 70 ms, which was expected to result in a positive and negative peak between 100 and 150 ms after averaging, resembling the complex P1/N1. Eventually, a phase desynchronization was induced at 170 ms to prevent the ERP from continuing indefinitely (Fig 5B). We also injected the data with some variability to emulate natural brain dynamics (such as pink noise amplitude and frequency drifts). Using this model, the scalp distribution was generated at 128 channels (same as our empirical data). We used the ‘ft_dipolesimulation’ function from Fieldtrip [31]. We validated this simulated dataset by computing the average for each condition (LVF and RVF) and plotting the ipsilateral and contralateral ERPs of the same posterior electrodes used for the empirical data analysis (that is, P5/P6, P7/P8, P9/P10, PO7/PO8, PO9/PO10, PO11/PO12, I1/I2, POI1/POI2 and O1/O2). Lastly, we followed the same procedure described in section 2.4.2 to generate its TGM from these posterior channels.

Fig 5. Simulated data. A. Dipole location. This image shows both the location of the two dipoles used for the simulations (left and right visual cortex) and the electrodes on which the dipole signal was projected. B. ERP simulated data. The dipole signal of the dipole was used to generate each experimental condition is shown over three trials. It starts with a random phase, which is interrupted by a stimulus-induced phase reset. C. Alpha simulated data. This panel depicts one trial of the two dipole signals used to generate each cue condition (red and blue line). Both dipoles are in phase and, after the onset of the presumed cue at 0 ms, the contralateral one shows a power decrease (blue line) while the ipsilateral one increases in power (red line).

The second simulated dataset was intended to mimic a contralateral decrease and ipsilateral increase in occipital alpha power in relation to the attended hemifield after a cue onset (Fig 5C). Thus, two oscillating dipoles were placed on the left and right hemifield with a frequency of 10 Hz and a random phase across trials. In each experimental condition (LVF and RVF), the contralateral dipole signal initiated a power decrease at 0.35 s, peaking at 0.5-0.7 ms and reaching previous values at 1.25 s, meanwhile the ipsilateral dipole had a mirrored increase. All other parameters remained the same as in the ERPs simulation. The validation of this dataset consisted in the estimation of the oscillatory power using a Hanning-tapered sliding window Fourier transform. The data was decomposed in 10 ms steps, obtaining frequencies from 4 to 30 Hz in 1 Hz steps and with the Hanning window width adjusted to 5 cycles per frequency. Then, we plotted the posterior contralateral and ipsilateral alpha (10 Hz) for each condition. Eventually, we computed the TGM, using the posterior channels as features (see 2.4.2 section).

The last step was to determine which of the two simulated TGM aligns most closely with the empirical TGM obtained for each dataset. We computed a 2-D correlation coefficient for each participant correlating their empirical TGM with the two simulated TGMs (ERP and alpha). This was done separately for each dataset, resulting in a single correlation value per participant, simulation and dataset. Correlation coefficients were normalized by applying a Fisher Z-transformation and their absolute values were statistically tested with a two-factor ANOVA, with Simulation (ERP vs Alpha) as within subjects factor and Attentional Cue (peripheral vs central) as a between-subject factor. Greenhouse-Geisser correction was considered in case of nonsphericity and Bonferroni was used for multiple comparison correction. We estimated effect size using the partial eta-square (ηp2) method. ”

Likewise, there is now a new heading in the results section (pages 26, 27, 28 and 29):

“3.6. Simulation results

Two simulated datasets with the electrophysiological correlates of attention (ERPs and alpha modulation) when attending to the LVF and RVF were generated to provide evidence on the TGM pattern resulting from that ground truth. In each dataset, the experimental conditions differed on the dipole location, which were placed in the right or/and left visual cortex (see Fig 5A).

The ERP simulation consisted of a single dipole oscillating at 10 Hz in the contralateral hemisphere to the simulated attentional cue, with a phase reset around 70 ms (see Fig 5B). The dipole orientation was reversed when moving the dipole to generate the other experimental condition. As expected, when computing the average across trials and conditions (LVF and RVF) to validate the dataset, two consecutive contralateral peaks were observed in the posterior electrodes (see Fig 10A, first row). Resembling the P1/N1 complex, the first component was positive and peaked at ≈100 ms, while the second one had a negative polarity and a ≈180 ms latency. Finally, the TGM computed on posterior channels showed a reverting pattern between 70 and 170 ms, coinciding with the ERP component latency (Fig 10A, second row). The higher accuracy values were found in the main diagonal, while the lowest ones spread off-diagonal, due to the P1/N1 polarity reversion between hemispheres and conditions.

Fig 10 Simulated data. A. ERP simulation. The first row depicts the average of the simulated data, for each cue condition and each posterior hemisphere. A contralateral P1/N1 complex can be observed (blue line), validating this data simulation. The second row is the TGM of this simulated dataset, revealing an early reverting pattern at the P1/N1 complex latency. B. Alpha simulation. In the first row, the 10 Hz power of the simulated data has been displayed, again for each cue and posterior hemisphere. The blue line shows the expected contralateral alpha decrease. The second row depicts the TGM, where no pattern can be observed.

Regarding the alpha simulation, both experimental conditions had one 10 Hz oscillating dipole in each hemisphere. There was a power modulation from 0.35 to 1.2 s (Fig 5B), with a power increase in the ipsilateral dipole (red line) and a power decrease in the contralateral one (blue line). This modulation can be observed when decomposing the simulated signal into frequencies: posterior contralateral channels showed a decrease at 10 Hz during the same latency (Fig 10B, first row). However, this neural signature was not informative for the classifier to be able to decode the two experimental conditions, as shown by the TGM with random accuracy values (Fig 10B, second row).

Eventually, we computed the correlation between the empirical and the simulated TGMs and applied a two-way ANOVA with Simulation (ERP or alpha) as a within-subjects factor and Attentional Cue (peripheral cue dataset and central-symbolic cue dataset) as a between-subject factor to assess which simulation best fits the empirical datasets. This test revealed significant main effects for Simulation (F1=8.46, p=.006, ηp2=0.15, β-1=.81) and Attentional Cue (F1=5.54, p=.023, ηp2=0.11, β-1=.64) and a significant interaction between both factors (F1=4.62, p=.037, ηp2=0.09, β-1=.56). Specifically, when considering the best fit for the peripheral cue dataset (see Fig 11, left side), higher correlations were observed with the ERP simulation (M=.030, SD=.005) than with the alpha simulation (M=.012, SD=.002, t(23)=2. 28, p<.001), but no differences were found in the central-symbolic cue dataset (see Fig 11, right side). Additionally, the coefficients obtained from correlating each dataset with the ERP simulation (see green data in Fig 11) were higher for the peripheral cue dataset (M=.030, SD=.005) than for the central-symbolic cue dataset (M=.014, SD=.004, t(47)=2.48, p=.017), while there were no differences in the alpha simulation (red data in Fig 11).

Fig 11 Empirical data and simulation alignment. After obtaining the 2-D correlation coefficient between each participant TGM for the peripheral and central cue datasets (left and right, respectively) and each simulated TGM (EPR - green and alpha - red), the absolute values of its Fisher z-transformation have been represented on this raincloud plot. Crosses represent outliers.”

These results have been integrated with the ideas previously discussed (page 33):

“Importantly, the simulated data provide insight into the debate regarding the underlying brain activity that leads to specific TGM patterns. Our ERP simulated data shows that a stimulus-induced phase reset, or in other words, an ERP, simultaneously in different hemispheres can be decoded by a classifier. This is our speculation regarding the empirical peripheral dataset, where both the ERP modulation and the simultaneous reverting TGM pattern can be observed, with a significant correlation (Figure 7). However, there is not such a reversion in the TGM of the central cue dataset, where we controlled for purely sensory-driven processes related to the cue onset. Thus, although the P1/N1 component is evident (Fig 8), the classifier result suggests that it must have different characteristics, as there is no successful decoding. However, the scenario in relation to alpha is simpler, whose simulation leads to the same results as both empirical data: the power modulation in alpha is not detectable by the classifier in order to distinguish both experimental conditions (see Figure 7 and 8). Indeed, our analysis of the fit between the simulated and empirical data points out that the peripheral cue aligns better with the ERP simulation compared to the alpha simulation, supporting the idea that ERP activity drives the observed pattern. Therefore, our simulations support the conclusions drawn by relating the electrophysiological correlates of attention to the classifier perform

---

## [Decision Letter · Decision Letter 1]

17 Feb 2025

Evidence for a constant occipital spotlight of attention using MVPA on EEG data

PONE-D-24-40710R1

Dear Dr. Melcon Martin,

We’re pleased to inform you that your manuscript has been judged scientifically suitable for publication and will be formally accepted for publication once it meets all outstanding technical requirements.

Kind regards,

Marie-Constance Corsi

Academic Editor

PLOS ONE

Reviewers' comments:

Reviewer's Responses to Questions

**Comments to the Author**

1. If the authors have adequately addressed your comments raised in a previous round of review and you feel that this manuscript is now acceptable for publication, you may indicate that here to bypass the “Comments to the Author” section, enter your conflict of interest statement in the “Confidential to Editor” section, and submit your "Accept" recommendation.

Reviewer #1: All comments have been addressed

2. Is the manuscript technically sound, and do the data support the conclusions?

Reviewer #1: Yes

3. Has the statistical analysis been performed appropriately and rigorously? 

Reviewer #1: Yes

4. Have the authors made all data underlying the findings in their manuscript fully available?

Reviewer #1: Yes

5. Is the manuscript presented in an intelligible fashion and written in standard English?

Reviewer #1: Yes

6. Review Comments to the Author

Reviewer #1: The authors have made a great job addressing my previous points. I am happy to recommend this paper for publication

7. PLOS authors have the option to publish the peer review history of their article (what does this mean? ). If published, this will include your full peer review and any attached files.

**Do you want your identity to be public for this peer review?** For information about this choice, including consent withdrawal, please see our Privacy Policy .

Reviewer #1: No

---

## [Editor Report · Acceptance letter]

PONE-D-24-40710R1

PLOS ONE

Dear Dr. Melcón,

I'm pleased to inform you that your manuscript has been deemed suitable for publication in PLOS ONE. Congratulations! Your manuscript is now being handed over to our production team.

Kind regards,

on behalf of

Dr. Marie-Constance Corsi

Academic Editor

PLOS ONE